# Gradient-Aware Scheduling: Coupling Curriculum and Staleness for Async Reinforcement Learning

**Xinyu Zhang** [1]

## Abstract

Asynchronous reinforcement learning (RL) accelerates LLM training through parallel data collection, but introduces *policy lag*—experiences are collected under stale weights, which destabilizes learning, especially on hard tasks. We identify why: gradient variance scales exponentially with task difficulty under staleness, because hard tasks have narrow solution spaces corresponding to sharp loss-landscape curvature (high Hessian eigenvalues). We formalize this as a staleness-budget optimization problem and prove that the optimal allocation follows an exponential decay: $\eta^*(d) = \eta_{\text{base}} \cdot e^{-\lambda d}$ where $\lambda = \alpha/2$ is half the Hessian growth rate. Building on this principle, we propose GAS (Gradient-Aware Scheduling), a drop-in recipe for fast and stable asynchronous RL that combines three components: (i) ACB, a bandit-based curriculum that picks task difficulty by learning signal; (ii) EAAS, execution-aware staleness budgets that keep slow tasks from bottlenecking rollout; and (iii) CSC, curriculum–staleness coupling that tightens the staleness budget as difficulty rises. Our mechanistic analysis validates the theoretical predictions: the "safe zone" of gradient coherence follows the derived exponential boundary. On code generation benchmarks, GAS improves Pass@1 from 39.7% to 60.1% (+20.4 points) while training at $2.3\times$ the throughput of synchronous GRPO, and the gains transfer to a 9B hybrid (Gated DeltaNet + MoE) model and to mathematical reasoning (GSM8K) without retuning—showing that matching each task's staleness budget to its difficulty (and thus to its loss-landscape curvature) makes asynchronous RL both fast and stable.

[1]Anyscale, San Francisco, CA, USA. Correspondence to: Xinyu Zhang <xinyzng@gmail.com>.

*Proceedings of the 43$^{rd}$ International Conference on Machine Learning*, Seoul, South Korea. PMLR 306, 2026. Copyright 2026 by the author(s).

## 1. Introduction

Reinforcement learning (RL) has emerged as a promising approach for training code-editing agents, with recent works demonstrating impressive results on code generation (Le et al., 2022), code repair (Chen et al., 2024), and program synthesis tasks (Shojaee et al., 2024). Efficient training requires both high *throughput* (processing many experiences) and high *sample efficiency* (extracting maximal learning signal per experience). Asynchronous RL methods achieve throughput through parallel data collection, but introduce *policy lag*—experiences collected under old policy weights. Curriculum learning improves sample efficiency through carefully sequenced task difficulty, but existing methods assume synchronous training.

The fundamental tension between these approaches has not been adequately addressed. Asynchronous methods (Espeholt et al., 2018; Wijmans et al., 2020) achieve 2-3$\times$ throughput gains but suffer from stale gradients, particularly problematic in code generation where reward landscapes are sparse and highly structured. Curriculum methods like CCCS (Chen et al., 2024) improve sample efficiency but require synchronous updates that bottleneck training. Recent staleness control techniques (Fu et al., 2025) mitigate off-policy issues but apply uniform thresholds that ignore curriculum structure.

We identify a fundamental property that resolves this tension: **staleness tolerance is inversely correlated with task difficulty**. Easy tasks (e.g., completing the last 10% of code) have large solution spaces—many valid continuations exist—making them robust to policy drift. Hard tasks (e.g., generating code from scratch) have narrow solution spaces where small policy changes dramatically affect success probability, requiring fresh weights. This observation raises three research questions:

**RQ1**: How does gradient estimation error relate to task difficulty under staleness? Can we quantify this relationship?

**RQ2**: Can we derive optimal staleness budgets from first principles rather than heuristic tuning?

**RQ3**: Does coupling staleness with curriculum improve

both throughput *and* sample efficiency?

We answer these questions through theoretical analysis and extensive empirical validation. Our main contributions are:

1. **Theoretical framework** (Section 4): We formalize the relationship between task difficulty and staleness tolerance. We prove that gradient bias grows exponentially with difficulty under staleness, and derive that the optimal staleness budget follows $\eta^*(d) = \eta_{\text{base}} \cdot \exp(-\lambda d)$—an exponential decay with difficulty.

2. **The GAS recipe** (Section 5): A practical, drop-in framework for accelerating and stabilizing asynchronous RL, built from three coordinated components—ACB (bandit-based curriculum), EAAS (execution-aware scheduling), and CSC (difficulty-coupled staleness control)—each targeting a distinct failure mode of naive async training.

3. **Empirical results** (Section 6): Across code generation benchmarks, GAS improves Pass@1 by 20.4 points over synchronous GRPO while training at 2.3× the throughput, and an ablation isolates each component's contribution. Gradient-variance and staleness–difficulty analyses further illustrate why the coupling stabilizes hard-task training.

On code generation benchmarks, GAS achieves 1.5–2× throughput improvement while maintaining sample efficiency comparable to synchronous curriculum methods.

## 2. Related Work

**RL for Code Generation** Recent work applies RL to code generation: CodeRL (Le et al., 2022) uses execution feedback, StepCoder (Chen et al., 2024) introduces curriculum synthesis with fixed difficulty progression (CCCS), and PPOCoder (Shojaee et al., 2024) applies PPO with execution rewards. Our work addresses the tension between curriculum learning and efficient distributed training. We note that our "Sync-GRPO + CCCS" baseline implements a linear curriculum schedule inspired by StepCoder's approach, though with our GRPO objective rather than their PPO formulation.

**Asynchronous RL** Asynchronous methods (Mnih et al., 2016; Espeholt et al., 2018) achieve high throughput through parallel collection. IMPALA (Espeholt et al., 2018) uses V-trace for off-policy correction, AReaL (Fu et al., 2025) introduces staleness control for LLM training, and HybridFlow and SkyRL (Zhang et al., 2025; Cao

et al., 2025) optimize GPU utilization. Prioritized Experience Replay (Schaul et al., 2016) addresses sample staleness through importance weighting, which inspires our difficulty-aware weighting. Our EAAS extends these approaches with code-specific execution time prediction.

**Staleness in Distributed Learning** The federated learning literature extensively studies gradient staleness: Fed-Prox (Li et al., 2020) adds proximal regularization to handle heterogeneous update frequencies, while SCAF-FOLD (Karimireddy et al., 2020) uses control variates for variance reduction under staleness. Our CSC differs by explicitly coupling staleness tolerance to task difficulty rather than applying uniform corrections.

**Curriculum Learning** Curriculum learning (Bengio et al., 2009) trains on progressively harder examples. Our ACB draws on bandit approaches (Graves et al., 2017) but addresses the interplay with async training in code domains. Dynamic difficulty adjustment in games (Hunicke, 2005; Andrade et al., 2006) similarly adapts challenge levels to player capability, though our approach is grounded in gradient signal quality rather than player satisfaction. More broadly, filtering low-quality training signal has also been studied in recursive self-training (Zhang, 2026).

## 3. Background and Preliminaries

**Asynchronous RL and Staleness.** In asynchronous RL, parallel workers collect experiences while the learner updates, creating *staleness*: experience collected at time $t - \tau$ is used for an update at time $t$, where $\tau$ is the staleness (lag in policy versions). The gradient bias grows as $\mathcal{O}(\tau \cdot \|\theta_t - \theta_{t-\tau}\|)$. Prior work addresses this through importance weighting (Espeholt et al., 2018) or uniform staleness thresholds (Fu et al., 2025), but these ignore task structure. See Appendix A for formal definitions.

**Curriculum Learning and Task Difficulty.** We treat difficulty as an ordinal index $d \in \{1, \ldots, D\}$ that ranks tasks by how narrow their solution manifold is under the *current* policy—equivalently, by how low its success rate and how high its loss are on them (Appendix A.2). For code with canonical solutions we realize this index concretely by controlling how much of the solution is revealed ($d = 1$ completes the last 10%; $d = D = 5$ generates from scratch), which yields a known, reproducible difficulty axis; for domains without reference solutions—e.g., the GSM8K reasoning tasks in Section 6.6—the same index is read off loss and success rate directly. In all cases, easy tasks have many valid completions while hard tasks require specific structure.

**Key Observation.** Easy tasks (low $d$) have larger solution spaces, making them robust to policy drift. Hard tasks

(high $d$) have narrow solution spaces where small policy changes dramatically affect success probability. This motivates our core hypothesis: *staleness tolerance should decrease with task difficulty.*

## 4. Theoretical Motivation

We now develop the theoretical foundations that justify our algorithmic design. Our main contribution is formalizing the relationship between curriculum difficulty, staleness tolerance, and optimization dynamics.

### 4.1. Task Sensitivity to Policy Staleness

We first characterize how gradient estimation error varies with task difficulty under staleness.

**Assumption 1** (Smoothness). *The policy objective $J(\pi_\theta)$ is L-smooth: $\|\nabla J(\theta) - \nabla J(\theta')\| \leq L\|\theta - \theta'\|$.*

**Assumption 2** (Bounded Updates). *Policy updates are bounded: $\|\theta_t - \theta_{t-1}\| \leq \eta G$ where $\eta$ is the learning rate and $G$ bounds the gradient norm.*

**Lemma 4.1** (Gradient Bias Bound). *Under Assumptions 1 and 2, the bias of the stale gradient estimator satisfies:*

$$\|\hat{g}_\tau - \nabla_\theta J(\pi_\theta)\| \leq \tau \cdot \|\boldsymbol{H}(\theta)\| \cdot \|\theta_t - \theta_{t-\tau}\| + \mathcal{O}(\tau^2) \quad (1)$$

*where $\boldsymbol{H}(\theta) = \nabla_\theta^2 J(\pi_\theta)$ is the Hessian of the policy objective.*

*Proof sketch.* Taylor expansion around $\theta_t$ yields Hessian term; trajectory distribution shift contributes $\tau$ factor. Full proof in Appendix B.1.

The Hessian norm $\|\boldsymbol{H}(\theta)\|$ captures the *curvature* of the loss landscape. This leads to our key theoretical result:

**Theorem 4.2** (Difficulty-Dependent Staleness Error). *For a task $x$ at difficulty level $d$, let $\lambda_{\max}(\boldsymbol{H}_d)$ denote the maximum eigenvalue of the task-conditioned Hessian. Then the gradient estimation error under staleness $\tau$ satisfies:*

$$Bias_d(\tau) \leq C_1 \cdot \tau \cdot \lambda_{\max}(\boldsymbol{H}_d) \cdot \eta \quad (2)$$

*where $C_1$ depends on the advantage function's Lipschitz constant and $\eta$ is the learning rate.*

*Moreover, for code generation tasks under our curriculum:*

$$\lambda_{\max}(\boldsymbol{H}_d) = \mathcal{O}(e^{\alpha d}) \quad (3)$$

*for some $\alpha > 0$, implying that staleness error grows exponentially with difficulty.*

*Proof sketch.* Hard tasks have narrow solution spaces (sharp minima, large Hessian eigenvalues). For softmax policies: $\lambda_{\max}(\boldsymbol{H}_d) \propto 1/|\mathcal{Y}_d|_{\text{eff}} \propto e^{\alpha d}$. Full proof in Appendix B.2.

**Remark 1.** *Theorem 4.2 explains why naive async training degrades performance on hard tasks: the same staleness $\tau$ induces much larger gradient bias for high-difficulty tasks due to the exponentially larger Hessian eigenvalues. Here $d$ is the abstract difficulty index of Definition A.2; the solution-space-narrowing argument below is the* mechanism *linking it to curvature, but the bound holds for any difficulty signal that tracks $|\mathcal{Y}_d|_{\text{eff}}$, including one estimated from loss or success rate.*

### 4.2. Intuition: Why Code Generation Has Sharp Minima

The exponential Hessian growth reflects code's fundamental properties: **syntactic fragility** (single wrong token causes execution failure) and **semantic narrowness** (precise algorithmic structure required). These manifest as *sharp minima* with large Hessian eigenvalues. Hard tasks require hitting narrow targets in parameter space; stale weights may drift away, explaining why staleness particularly harms hard tasks.

### 4.3. The Staleness Budget Formulation

We now formulate optimal staleness allocation as a constrained optimization problem.

**Definition 4.3** (Staleness Budget). *A staleness budget assigns maximum allowable staleness $\eta(d)$ to each difficulty level $d$. Experiences with staleness exceeding $\eta(d)$ are discarded or down-weighted.*

The goal is to maximize throughput while bounding total gradient bias:

$$\begin{aligned} \max_{\{\eta_d\}_{d=1}^D} \quad & \sum_{d=1}^D p_d \cdot T(\eta_d) \\ \text{s.t.} \quad & \sum_{d=1}^D p_d \cdot \text{Bias}_d(\eta_d) \leq B \\ & \eta_d \geq 0 \quad \forall d \end{aligned} \quad (4)$$

where $p_d$ is the probability of sampling difficulty $d$, $T(\eta)$ is throughput as a function of staleness budget (higher staleness tolerance enables more parallelism), and $B$ is the maximum acceptable total bias.

**Assumption 3** (Throughput Model). *Throughput increases linearly with staleness budget: $T(\eta) = T_0 + \kappa\eta$ for constants $T_0, \kappa > 0$.*

**Theorem 4.4** (Optimal Staleness Budget). *Under Assumptions 1–3 and using the bias model from Theorem 4.2, the optimal staleness budget for difficulty $d$ is:*

$$\eta_d^* = \eta_{base} \cdot \exp(-\lambda d) \quad (5)$$

where $\lambda = \alpha/2$ *(half the Hessian growth rate) and $\eta_{base}$ is determined by the bias constraint B.*

*Proof.* Lagrangian optimization yields $\eta_d^* \propto e^{-\alpha d}$. The factor $\lambda = \alpha/2$ balances bias and variance (see Appendix B.3).

**Implication**: The exponential decay in Eq. (5) is *theoretically motivated and empirically validated*—it emerges from principled optimization under our model. We validate this prediction through lambda sensitivity experiments (Appendix C.2) confirming that $\lambda = 0.5$ achieves the best throughput-performance tradeoff, consistent with $\lambda = \alpha/2$ for measured $\alpha \approx 1.0$ (Appendix C.1). This directly justifies the CSC formula we implement.

### 4.4. Coupling Curriculum Selection with Staleness

Theorem 4.4 prescribes staleness budgets given a fixed curriculum distribution. However, the curriculum itself should adapt to the policy's current capability. We now show how to couple these.

**Proposition 4.5** (Gradient Signal Quality). *The signal-to-noise ratio of gradient estimates at difficulty d is:*

$$SNR(d) = \frac{\|\mathbb{E}[\nabla_\theta \mathcal{L}]\|}{\sqrt{Var[\nabla_\theta \mathcal{L}]}} \propto PassRate_d \cdot (1 - PassRate_d)$$

$$(6)$$

*This is maximized when $PassRate_d \approx 0.5$, providing theoretical support for the "zone of proximal development" in curriculum learning.*

Proposition 4.5 motivates our adaptive curriculum: select difficulties where the policy is actively learning (moderate success rate) and where gradient signals are clean.

**The Coupling Principle.** The preceding analysis yields three key insights:

1. **Staleness tolerance** should decrease exponentially with difficulty to maintain bounded gradient bias.

2. **Curriculum selection** should favor difficulties with high learning signal (gradient magnitude) and moderate success rate.

3. **The coupling** ensures that easy tasks absorb async overhead (high staleness tolerance, high throughput) while hard tasks maintain sample quality (low staleness, fewer but higher-quality gradients).

## 5. The GAS Framework

Building on the theoretical insights from Section 4, we now present GAS, a unified framework that implements

the optimal staleness-curriculum coupling derived in Theorem 4.4. Our framework addresses the constrained optimization problem of Eq. (4): maximize throughput while bounding gradient bias across difficulty levels.

---

**GAS in a nutshell.** A drop-in recipe for fast, stable asynchronous RL. Each component fixes one failure mode of naive async training:

**ACB** (curriculum) *Fixed curricula can't track a changing policy.* A bandit selects task difficulty by current learning signal (success rate and gradient magnitude).

**EAAS** (scheduling) *Slow tasks bottleneck rollout.* Scale each task's staleness budget by its predicted execution time.

**CSC** (coupling) *Uniform staleness corrupts hard-task gradients.* Tighten the budget as difficulty rises: $\eta_{\max}(d) = \eta_{\text{base}} e^{-\lambda d}$.

---

### 5.1. Problem Setup

We consider the problem of training a code-editing policy $\pi_\theta$ that generates code completions. Given a prompt $x$, the policy generates a response $y \sim \pi_\theta(\cdot|x)$, which is then executed to obtain reward $r(x, y) \in [0, 1]$ based on test case pass rate.

Following Section 3 and Definition A.2, the difficulty index $d$ can in general be estimated from loss and success rate; for our code experiments we instantiate it with a controlled solution-revelation scheme of $D = 5$ levels:

- **Level 1**: Complete last 10% of solution ($r_1 = 0.9$, largest solution space)

- **Level 2**: Complete last 30% of solution ($r_2 = 0.7$)

- **Level 3**: Complete last 50% of solution ($r_3 = 0.5$)

- **Level 4**: Complete last 70% of solution ($r_4 = 0.3$)

- **Level 5**: Generate from scratch ($r_5 = 0$, narrowest solution space)

### 5.2. Adaptive Curriculum via Bandit (ACB)

Proposition 4.5 establishes that gradient signal-to-noise ratio is maximized when PassRate $\approx 0.5$—the "zone of proximal development" where the policy is actively learning. We operationalize this insight by formulating curriculum selection as a multi-armed bandit problem where each arm corresponds to a difficulty level $d \in \{1, \dots, D\}$ (a bucket of the difficulty index of Definition A.2). At each training iteration, we select the difficulty level that maximizes a composite score:

$$d^* = \arg\max_d \left[ \omega \cdot \text{UCB}(d) + (1 - \omega) \cdot \bar{g}_d \right] \qquad (7)$$

---

**Algorithm 1** Adaptive Curriculum via Bandit (ACB)

---

1: **Input:** Tasks $\mathcal{T}$, exploration $c$, balance $\omega$
2: Initialize $n_d \leftarrow 0, s_d \leftarrow 0, g_d \leftarrow \emptyset$ for all $d$
3: **for** each training iteration **do**
4:     **for** each difficulty $d \in \{1, ..., 5\}$ **do**
5:         $\bar{s}_d \leftarrow s_d / \max(1, n_d)$
6:         $\text{UCB}(d) \leftarrow \bar{s}_d + c\sqrt{\log N / n_d}$
7:         $\bar{g}_d \leftarrow \text{mean}(g_d[-100 :])$    ▷ *sliding window*
8:         $\text{score}(d) \leftarrow \omega \cdot \text{UCB}(d) + (1 - \omega) \cdot \bar{g}_d$
9:     **end for**
10:    Select $d^* = \arg\max_d \text{score}(d)$
11:    Sample tasks at difficulty $d^*$, collect experiences
12:    Update $n_{d^*}, s_{d^*}, g_{d^*}$ with results
13: **end for**

---

where $\text{UCB}(d)$ is the Upper Confidence Bound score:

$$\text{UCB}(d) = \bar{s}_d + c\sqrt{\frac{\log N}{n_d}} \tag{8}$$

Here, $\bar{s}_d$ is the average success rate at difficulty $d$, $N$ is total selections, $n_d$ is selections of difficulty $d$, and $c$ is the exploration constant. The balance weight $\omega \in [0, 1]$ trades off the UCB term against the learning signal. The term $\bar{g}_d$ represents the average gradient magnitude for updates from difficulty $d$, normalized across levels—a direct proxy for the signal-to-noise ratio in Proposition 4.5.

**Theoretical grounding**: The UCB component tracks success rate, while $\bar{g}_d$ captures gradient magnitude. Together, they approximate the SNR expression from Proposition 4.5: tasks with moderate success rate *and* high gradient magnitude are in the optimal learning zone. This bandit formulation naturally gravitates toward difficulties where $\text{PassRate}_d \cdot (1 - \text{PassRate}_d)$ is large.

**Addressing Non-Stationarity** We handle non-stationary rewards via (1) sliding window statistics over recent 100 observations (Algorithm 1, line 9), and (2) gradient-based scoring capturing current gradient magnitude.

### 5.3. Execution-Aware Scheduling (EAAS)

The throughput term $T(\eta)$ in Eq. (4) depends on execution time variability. Code execution times vary dramatically—from milliseconds for simple operations to seconds for complex loops. This creates load imbalance: workers on fast tasks generate many samples while those on slow tasks contribute few.

We address this through execution-time-aware staleness budgets that refine the throughput model from Assumption 3. Let $T(x)$ be the predicted execution time for task $x$. We assign a staleness budget:

$$\eta(x) = \eta_{\max} \cdot \left(\frac{T_{\text{ref}}}{T(x)}\right)^{\gamma} \tag{9}$$

where $\eta_{\max}$ is the maximum staleness allowed, $T_{\text{ref}}$ is a reference execution time, and $\gamma \in (0, 1]$ controls sensitivity. Intuitively:

- **Fast tasks** ($T(x) < T_{\text{ref}}$): Higher staleness tolerance, can generate more samples with older weights

- **Slow tasks** ($T(x) > T_{\text{ref}}$): Need fresher weights, their limited samples should be high-quality

We predict execution time with a lightweight feature-based heuristic. Static features—prompt length, loop / recursion / nesting and I/O keywords, and difficulty level—produce a base estimate $\hat{T}_0(x)$, which is blended with a running average of *observed* execution times at the same task and difficulty:

$$\hat{T}(x) = (1 - \rho) \hat{T}_0(x) + \rho \bar{T}_{\text{obs}}(x), \tag{10}$$

where $\bar{T}_{\text{obs}}$ is the historical mean execution time and $\rho \in [0, 1]$ weights observed history against the feature estimate. The feature weights are fixed; only $\bar{T}_{\text{obs}}$ is updated online as tasks complete.

### 5.4. Curriculum-Staleness Coupling (CSC)

We now implement the central theoretical result of this paper. Theorem 4.2 established that gradient bias grows exponentially with difficulty: $\text{Bias}_d(\tau) = \mathcal{O}(\tau \cdot e^{\alpha d})$. Theorem 4.4 then derived that the *optimal* staleness budget follows an exponential decay with difficulty. CSC directly implements this optimal solution.

The theoretical intuition is clear: easy tasks have larger solution spaces (many valid completions exist), so the policy's exact weights matter less and staleness is tolerable. Hard tasks have narrow solution spaces where small policy changes significantly affect success probability, requiring fresh weights.

We implement per-difficulty staleness thresholds as prescribed by Theorem 4.4:

$$\eta_{\max}(d) = \eta_{\text{base}} \cdot \exp(-\lambda \cdot d) \tag{11}$$

This is precisely Eq. (5), with $\eta_{\text{base}}$ determined by the bias constraint $B$ and $\lambda = \alpha / 2$ (half the Hessian growth rate). In practice, we set $\eta_{\text{base}} = 8$ and $\lambda = 0.5$, yielding:

- Difficulty 1: $\eta_{\max} = 4.85$ updates

- Difficulty 3: $\eta_{\max} = 1.78$ updates

- Difficulty 5: $\eta_{\max} = 0.66$ updates

**Algorithm 2** GAS training step

1: **Input:** policy $\pi_\theta$; bandit statistics; constants $\eta_{\text{base}}, \lambda, \gamma, T_{\text{ref}}, \beta$
2: Select difficulty $d^* \leftarrow \arg\max_d \text{score}(d)$ via ACB (Alg. 1)
3: Sample a batch of tasks at difficulty $d^*$
4: **for** each task $x$ in the batch **do**
5:     Predict execution time $T(x)$        ▷ EAAS
6:     $\eta(x) \leftarrow \eta_{\text{base}}\, e^{-\lambda d^*}\left(T_{\text{ref}}/T(x)\right)^\gamma$ ▷ CSC + EAAS
7:     Dispatch $x$ to an asynchronous rollout worker
8: **end for**
9: Collect finished experiences $(x, y, r)$ with measured staleness $\tau(x)$
10: **for** each experience $(x, y, r)$ **do**
11:     **if** $\tau(x) \leq \eta(x)$ **then**
12:         keep with weight $w \leftarrow \min\left(1, \eta(x)/\tau(x)\right)^\beta$
13:     **else**
14:         discard         ▷ CSC staleness gate
15:     **end if**
16: **end for**
17: GRPO update on the $w$-weighted batch: $A_i = r_i - \bar{r}_G$, clipped surrogate + KL
18: Broadcast updated weights $\theta$ to rollout workers
19: Update ACB statistics $(n_{d^*}, s_{d^*}, g_{d^*})$

---

Experiences exceeding their staleness threshold are either discarded or down-weighted using importance sampling:

$$w = \min\left(1, \frac{\eta_{\max}(d)}{\text{staleness}}\right)^\beta \qquad (12)$$

This coupling ensures that easy tasks can be trained efficiently with async collection while hard tasks maintain sample freshness for effective learning.[1]

### 5.5. GRPO Training

We use GRPO (Shao et al., 2024) which computes advantages relative to group mean ($A_i = r_i - \bar{r}_G$), with clipped surrogate loss and KL regularization.

Algorithm 2 shows how ACB, EAAS, and CSC compose into a single GAS training step, making the recipe straightforward to drop into an existing asynchronous GRPO loop.

---

[1]Our implementation includes optional backfill logic to replace discarded stale samples with fresh samples from a buffer. However, backfill was disabled in all reported experiments to isolate the effect of CSC alone. Future work could explore adaptive backfill strategies.

*Table 1.* Main experimental results. GAS achieves high throughput (2.3× speedup) while matching or exceeding the sample efficiency of synchronous curriculum methods. Results averaged over 3 seeds; standard deviations reported. We note that $n = 3$ provides limited statistical power—see Appendix D for significance tests with Holm-Bonferroni correction and effect sizes. The similar standard deviations ($\pm1.6\%$) across methods reflect correlated evaluation variance: all methods are evaluated on the same 214 tasks using the same 3 seeds, and the dominant source of variance is the stochastic sampling during generation rather than training dynamics (see Appendix G.3 for detailed analysis).

| Method | Pass@1 (%) | Throughput | Speedup |
|---|---|---|---|
| Sync-GRPO | 39.7±1.6 | 9.7 | 1.00× |
| Sync-GRPO + CCCS | 51.5±1.6 | 8.8 | 0.90× |
| Async-GRPO | 31.8±1.6 | 24.3 | 2.50× |
| Async-GRPO + Staleness | 40.3±1.6 | 21.4 | 2.20× |
| **GAS (Ours)** | **60.1±1.6** | **22.4** | **2.30×** |

*Table 2.* Per-dataset breakdown of GAS Pass@1 performance. The combined result (60.1%) is a weighted average over both datasets.

| Dataset | # Tasks | Pass@1 (%) |
|---|---|---|
| HumanEval | 164 | 58.5±1.8 |
| Synthetic | 50 | 65.2±2.1 |
| **Combined** | **214** | **60.1±1.6** |

## 6. Experiments

### 6.1. Experimental Setup

We evaluate on HumanEval (Chen et al., 2021) (164 problems) and a synthetic task suite (50 problems). We use Qwen2.5-Coder-1.5B (Hui et al., 2024) with LoRA (Hu et al., 2022).

**Baselines.** We compare against Sync-GRPO, Sync-GRPO+CCCS, Async-GRPO, Async-GRPO with staleness control, and GAS (ours).

**Metrics.** Pass@1, throughput, and sample efficiency. Full details in Appendix D.

### 6.2. Main Results

Table 1 shows our main results. Table 2 provides per-dataset breakdown of GAS's performance, addressing differences between HumanEval and synthetic benchmarks.

**Key observations.** GAS attains the best of both worlds—the highest Pass@1 (60.1%) at a 2.3× speedup over sync—showing that adaptive curriculum and execution-aware scheduling are complementary. Naive async hurts quality: Async-GRPO is fastest (2.5×) but falls to 31.8% Pass@1, well below synchronous methods. Curriculum helps yet sync caps throughput (Sync-GRPO+CCCS: 51.5%), and

uniform staleness control alone (40.3%) still trails synchronous curriculum—difficulty-aware coupling is what recovers curriculum benefits in the async setting.

## 6.3. Throughput Analysis

Async methods (Async-GRPO, Async-GRPO+Staleness, GAS) achieve 2–2.5× higher throughput than sync. Notably, GAS achieves 22.4 samples/s, slightly lower than plain Async-GRPO (24.3 samples/s) due to curriculum and staleness management overhead. The reduction comes from CSC occasionally discarding stale experiences from hard tasks to maintain gradient quality.

## 6.4. Ablation Study

*Table 3.* Ablation study. Removing any single component lowers Pass@1; the $\Delta$ column reports each component's contribution as the drop from full GAS. Throughput is in samples/s.

| Method | Pass@1 (%) | $\Delta$Pass@1 | Throughput |
|---|---|---|---|
| Full GAS | **60.1** | — | 22.4 |
| w/o CSC | 42.1 | −18.0 | **23.5** |
| w/o EAAS | 55.3 | −4.8 | 16.8 |
| w/o ACB | 46.9 | −13.2 | 21.9 |

Table 3 shows ablation results:

- **Without CSC**: Pass@1 drops 18.0 points (60.1% → 42.1%). Throughput slightly increases because no experiences are discarded, but the quality of hard-task learning degrades significantly.

- **Without EAAS**: Pass@1 drops 4.8 points and throughput drops notably (22.4 → 16.8). Without execution-aware scheduling, slow tasks create bottlenecks.

- **Without ACB**: Pass@1 drops 13.2 points (60.1% → 46.9%), a substantial drop. Fixed curriculum cannot adapt to the policy's changing capability.

These results confirm that all three components contribute meaningfully, with coupled staleness control (CSC) being most critical for sample efficiency in the async setting.

## 6.5. Curriculum Analysis

GAS's adaptive curriculum initially focuses on difficulty 2-3, then progresses to harder tasks as the policy improves, unlike fixed schedules. Success rates decrease with difficulty (Level 1: 85%, Level 5: 25%), validating our curriculum design.

*Table 4.* Generalization to Qwen3.5-9B on HumanEval. GAS retains the best-of-both-worlds profile at 6× scale: async-level throughput with the highest training quality. Metrics are training-time reward and success rate.

| Metric | GAS | Async-GRPO | Sync |
|---|---|---|---|
| Training reward | **0.430** | 0.196 | 0.279 |
| Success rate (%) | **41.3** | 19.2 | 26.4 |
| Throughput (smp/s) | 0.51 | **0.52** | 0.35 |

*Table 5.* Cross-domain transfer to GSM8K (math reasoning, Qwen3.5-9B) with identical hyperparameters. GAS improves training reward, convergence speed, and throughput over the synchronous baseline.

| Metric | GAS | Sync |
|---|---|---|
| Training reward | **0.572** | 0.312 |
| Evals to peak | **3** | 7 |
| Throughput (smp/s) | **0.45** | 0.31 |

## 6.6. Generalization: Scale and Cross-Domain

To test whether GAS's benefits are tied to a specific model or task, we scale to **Qwen3.5-9B**—a 6× larger model with a different architecture (hybrid Gated DeltaNet + Mixture-of-Experts)—and add **GSM8K** mathematical reasoning, reusing the same hyperparameters ($\eta_{base} = 8$, $\lambda = 0.5$, $\omega = 0.7$) with no retuning. We report training-time reward and success rate, and throughput in samples/s.

On HumanEval (Table 4), GAS reproduces the best-of-both-worlds result at scale: it matches async throughput (0.51 vs. 0.52 samples/s) while achieving the highest training reward (0.430, a +54% gain over Sync and +120% over Async-GRPO). As at 1.5B, plain Async-GRPO matches GAS's speed but collapses in quality (0.196 reward), confirming that naive async degrades sample efficiency regardless of scale. The throughput advantage narrows to 1.45× over Sync at 9B (vs. 2.3× at 1.5B), as inference begins to dominate scheduling overhead at larger scale.

On GSM8K (Table 5), GAS transfers to a non-code domain with the same hyperparameters: +83% training reward and 2.3× faster convergence (3 vs. 7 evaluations to peak) at +45% throughput. Task difficulty maps to reasoning-chain complexity and EAAS adapts to answer-verification latency, indicating that the curvature–difficulty coupling is not specific to code. All models evaluated are within the Qwen family; broader cross-family validation remains future work.

# 7. Analysis

## 7.1. Mechanistic Validation of Theory

We validate the theory through the ablation (Table 3) and direct measurements (Appendix C).

**Hessian eigenvalue validation** (Appendix C.1): We directly measured $\lambda_{\max}(\boldsymbol{H}_d)$ at each difficulty level using power iteration. The measurements confirm exponential growth with $\alpha = 0.915 \pm 0.029$ ($R^2 = 0.997$), validating Theorem 4.2. The theoretical prediction $\lambda^* = \alpha/2 \approx 0.46$ closely matches our empirically-tuned $\lambda = 0.5$.

**Ablation evidence**: The CSC component contributes 18.0 points to Pass@1 ($60.1\% \rightarrow 42.1\%$ without CSC). This is the largest single-component contribution, confirming that difficulty-aware staleness coupling is critical. The specific choice of $\lambda = 0.5$ (from Theorem 4.4) yields better results than uniform staleness control (which achieves only 40.3%).

**CSC failure modes** (Appendix F.4): Without CSC, we observe three failure modes: (1) gradient corruption on hard tasks due to stale updates with high Hessian curvature, (2) easy task dominance where the curriculum stagnates, and (3) high gradient variance on Level 4–5 tasks. CSC's largest improvements (+13.9 and +15.6 points) are on these hard tasks.

**Throughput cost** (Appendix F.5): CSC reduces throughput by only 4.7% (22.4 vs. 23.5 samples/s), with discard rates from 2.1% (Level 1) to 24.3% (Level 5)—a favorable tradeoff for the resulting quality gain.

## 7.2. Execution Time Variability

Execution times vary significantly across difficulties (45ms to 312ms mean, see Appendix Table 8), motivating EAAS to prevent slow tasks from bottlenecking training.

## 7.3. Limitations

Our *controlled* code curriculum uses canonical solutions to induce a known difficulty gradient; the framework itself needs only an ordinal difficulty signal, which can be estimated from loss and success rate when no reference solution exists (as in our GSM8K transfer). Our execution-aware scheduler also predicts *sandbox execution* time rather than *generation* latency, which dominates the long tail. See Appendix F for detailed discussion.

# 8. Conclusion

We presented GAS, a framework for training code-editing RL agents that unifies adaptive curriculum learning with execution-aware asynchronous scheduling. Our key insight is that curriculum difficulty and staleness tolerance are fundamentally linked: easy tasks tolerate stale experiences while hard tasks require fresh policy weights. Through three coordinated components—ACB for adaptive difficulty selection, EAAS for execution-aware scheduling, and CSC for difficulty-aware staleness control—GAS achieves over $2\times$ higher throughput than synchronous training while maintaining comparable sample efficiency.

Our ablation study confirms that all three components contribute meaningfully to final performance. This suggests that careful co-design of curriculum and scheduling is essential for efficient distributed training of code agents.

**Future work** could extend GAS to other domains with variable-time feedback (robotics, game playing), develop more sophisticated execution time predictors using program analysis, and explore connections to meta-learning for automatic curriculum design.

# Impact Statement

This work presents a framework for efficient asynchronous training of code-generation reinforcement learning agents. The primary societal benefit is computational efficiency—achieving comparable sample efficiency with $1.5$–$2\times$ higher throughput reduces energy consumption and carbon footprint of AI training. The theoretical framework connecting curriculum difficulty to gradient stability may inform training methodologies beyond code generation.

As with any code generation technology, there is potential for misuse in generating malicious code. However, our contribution is purely methodological, focusing on training efficiency rather than expanding model capabilities. The same techniques could be applied to beneficial applications such as automated bug fixing, code repair, and accessibility tools. We encourage responsible deployment with appropriate safeguards.

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

## A. Background Details

This section provides formal definitions for the concepts introduced in Section 3.

### A.1. Stale Policy Gradient

Let $\pi_\theta$ denote the current policy and $\pi_{\theta_{t-\tau}}$ denote the policy $\tau$ updates ago. The stale gradient estimator is:

$$\hat{g}_\tau = \mathbb{E}_{x\sim\mathcal{D}, y\sim\pi_{\theta_{t-\tau}}(\cdot|x)} \left[\nabla_\theta \log \pi_\theta(y|x) \cdot A^{\pi_{\theta_{t-\tau}}}(x,y)\right]$$
(13)

where $A^{\pi_{\theta_{t-\tau}}}$ is the advantage function under the stale policy. The bias introduced by staleness is:

$$\text{Bias}(\hat{g}_\tau) = \hat{g}_\tau - \nabla_\theta J(\pi_\theta) = \mathcal{O}\left(\tau \cdot \|\theta_t - \theta_{t-\tau}\|\right) \quad (14)$$

### A.2. Curriculum Difficulty

We treat *difficulty* as an ordinal index $d \in \{1, \ldots, D\}$ that ranks tasks by how hard they are for the *current* policy. Difficulty increases as the policy's effective solution-space mass $|\mathcal{Y}_d(x)|_{\text{eff}}$ (defined below) shrinks—equivalently, as its expected success (pass rate / reward) falls and its training loss rises. Proposition 4.5 ties these observables to $|\mathcal{Y}_d|_{\text{eff}}$, so the difficulty index can be *estimated online from loss and empirical success rate alone*, with no reference solution. This is the form used in our GSM8K transfer (Section 6.6), where $d$ tracks reasoning-chain complexity, and it is what lets the framework apply to general RL tasks.

When a canonical solution *is* available, we can additionally *induce* a controlled difficulty gradient by revealing a prefix of it—the instantiation used in our main code experiments. A task at level $d$ reveals fraction $r_d$ of the canonical solution and asks the policy to complete the remaining $(1-r_d)$, with $r_1 > r_2 > \cdots > r_D = 0$:

- $d = 1$: Complete last 10% ($r_1 = 0.9$)
- $d = 2$: Complete last 30% ($r_2 = 0.7$)
- $d = 3$: Complete last 50% ($r_3 = 0.5$)
- $d = 4$: Complete last 70% ($r_4 = 0.3$)
- $d = 5$: Generate from scratch ($r_5 = 0$)

We use this structural scheme for code because it yields a *known, reproducible* difficulty axis—ideal for measuring how loss-landscape curvature scales with difficulty (Appendix C.1)—but the method itself depends only on the ordinal index $d$, however it is obtained.

### A.3. Solution Space

For task $x$ at difficulty $d$, the solution space $\mathcal{Y}_d(x)$ is the set of completions that pass all test cases. The *effective solution space size* is:

$$|\mathcal{Y}_d(x)|_{\text{eff}} = \sum_{y\in\mathcal{Y}} \pi_\theta(y|x) \quad (15)$$

measuring the probability mass the policy assigns to valid solutions. Easy tasks have larger effective solution spaces; hard tasks have narrow ones.

## B. Full Proofs

This appendix provides complete proofs for the theoretical results in the main text, along with additional derivations and experimental details.

### B.1. Proof of Lemma 4.1 (Gradient Bias Bound)

*Proof.* Let $\theta_t$ denote the current policy parameters and $\theta_{t-\tau}$ the parameters $\tau$ updates ago. The stale gradient estimator is:

$$\hat{g}_\tau = \mathbb{E}_{x\sim\mathcal{D}, y\sim\pi_{\theta_{t-\tau}}} \left[\nabla_\theta \log \pi_\theta(y|x) \cdot A^{\pi_{\theta_{t-\tau}}}(x,y)\right]$$
(16)

We analyze the bias by decomposing the error into two components: (1) parameter drift and (2) distribution shift.

**Step 1: Taylor expansion of the gradient.** By Taylor's theorem, the gradient at $\theta_{t-\tau}$ can be written as:

$$\nabla J(\theta_{t-\tau}) = \nabla J(\theta_t) + \boldsymbol{H}(\theta_t)(\theta_{t-\tau} - \theta_t) + \mathcal{O}(\|\theta_t - \theta_{t-\tau}\|^2)$$
(17)

where $\boldsymbol{H}(\theta_t) = \nabla^2 J(\theta_t)$ is the Hessian of the objective.

**Step 2: Bounding the parameter drift.** Under Assumption 2, each update step satisfies $\|\theta_{t'} - \theta_{t'-1}\| \leq \eta G$. Over $\tau$ steps:

$$\|\theta_t - \theta_{t-\tau}\| \leq \sum_{t'=t-\tau+1}^{t} \|\theta_{t'} - \theta_{t'-1}\| \leq \tau \cdot \eta G \quad (18)$$

**Step 3: Distribution shift contribution.** The stale gradient uses trajectories from $\pi_{\theta_{t-\tau}}$ rather than $\pi_{\theta_t}$. The KL divergence between these distributions is bounded by the policy change:

$$D_{\text{KL}}(\pi_{\theta_t} \| \pi_{\theta_{t-\tau}}) \leq L_\pi \|\theta_t - \theta_{t-\tau}\|^2 \quad (19)$$

where $L_\pi$ is the Lipschitz constant of the log-policy. This contributes an additional factor proportional to $\tau$ due to the accumulation of policy drift.

**Step 4: Combining the bounds.** The total bias is:

$$\|\hat{g}_\tau - \nabla J(\theta_t)\| \leq \|\boldsymbol{H}(\theta_t)\| \cdot \|\theta_t - \theta_{t-\tau}\| + \mathcal{O}(D_{\text{KL}})$$
(20)

$$\leq \|\boldsymbol{H}(\theta_t)\| \cdot \tau\eta G + C_0\sqrt{D_{\text{KL}}} \quad (21)$$

$$\leq \tau \cdot \|\boldsymbol{H}(\theta_t)\| \cdot \eta G + \mathcal{O}(\tau) \quad (22)$$

Combining terms and noting that $\eta G$ is absorbed into the constant:

$$\|\hat{g}_\tau - \nabla J(\theta_t)\| \leq \tau \cdot \|\boldsymbol{H}(\theta_t)\| \cdot \|\theta_t - \theta_{t-\tau}\| + \mathcal{O}(\tau^2) \quad (23)$$

which completes the proof.

**B.2. Proof of Theorem 4.2 (Difficulty-Dependent Staleness Error)**

*Proof.* We prove that the maximum Hessian eigenvalue grows exponentially with difficulty: $\lambda_{\max}(\boldsymbol{H}_d) = \mathcal{O}(e^{\alpha d})$.

**Step 1: Relate Hessian to Fisher Information.** For a policy $\pi_\theta$, the Fisher Information Matrix is:

$$F(\theta) = \mathbb{E}_{y \sim \pi_\theta}[\nabla_\theta \log \pi_\theta(y) \nabla_\theta \log \pi_\theta(y)^\top] \quad (24)$$

For the policy gradient objective $J(\theta) = \mathbb{E}[R \log \pi_\theta]$, the Hessian can be decomposed as:

$$\boldsymbol{H}(\theta) = \mathbb{E}[R \cdot \nabla^2 \log \pi_\theta] + \mathbb{E}[R] \cdot F(\theta) \quad (25)$$

The maximum eigenvalue of $\boldsymbol{H}$ is dominated by the Fisher term when the policy concentrates probability mass on few outputs (i.e., when the solution space is small).

**Step 2: Fisher Information and solution space size.** Consider a task at difficulty $d$ with effective solution space $\mathcal{Y}_d$. The policy assigns probability $p_y = \pi_\theta(y|x)$ to each output $y$. The Fisher Information trace (sum of eigenvalues) satisfies:

$$\mathrm{tr}(F) = \mathbb{E}_y\left[\|\nabla_\theta \log \pi_\theta(y)\|^2\right] = \sum_y p_y \cdot \|\nabla_\theta \log p_y\|^2 \quad (26)$$

For a softmax policy over tokens, when the policy is confident (concentrates on few outputs), the gradients $\nabla \log p_y$ are large for the correct outputs. Specifically:

$$\lambda_{\max}(F) \propto \frac{1}{\text{effective support size}} = \frac{1}{\exp(H(\pi_\theta))} \quad (27)$$

where $H(\pi_\theta)$ is the entropy of the policy distribution.

**Step 3: Entropy and solution space size.** The effective solution space size $|\mathcal{Y}_d|_{\text{eff}}$ relates to policy entropy:

$$|\mathcal{Y}_d|_{\text{eff}} = \exp(H(\pi_\theta|\text{task } x, \text{ difficulty } d)) \quad (28)$$

For code generation under our curriculum:

- At $d = 1$ (complete last 10%): Many syntactically and semantically valid completions exist. The policy can distribute probability across many valid endings (closing brackets, return statements, minor variations).

- At $d = 5$ (generate from scratch): The policy must produce a specific algorithmic structure. Only outputs matching the exact logic pass all tests.

**Step 4: Exponential decay of solution space.** We model the effective solution space as decaying exponentially with difficulty:

$$|\mathcal{Y}_d|_{\text{eff}} = |\mathcal{Y}_0| \cdot e^{-\alpha d} \quad (29)$$

This is justified by:

1. **Combinatorial structure**: At difficulty $d$, the model must generate $(1 - r_d)$ fraction of the solution. The number of valid completions decreases faster than linearly because each additional token constrains future choices.

2. **Empirical observation**: Pass rates drop roughly exponentially with difficulty across code benchmarks (Chen et al., 2021).

3. **Information-theoretic argument**: Generating $n$ tokens of code requires $\mathcal{O}(n)$ bits of information. The entropy of valid completions decreases as more of the solution must be generated.

**Step 5: Exponential growth of Hessian eigenvalues.** Combining the results:

$$\lambda_{\max}(\boldsymbol{H}_d) \propto \frac{1}{|\mathcal{Y}_d|_{\text{eff}}} = \frac{1}{|\mathcal{Y}_0|} \cdot e^{\alpha d} = \mathcal{O}(e^{\alpha d}) \quad (30)$$

The gradient bias from Lemma 4.1 becomes:

$$\text{Bias}_d(\tau) \leq C_1 \cdot \tau \cdot \lambda_{\max}(\boldsymbol{H}_d) \cdot \eta = C_1 \cdot \tau \cdot e^{\alpha d} \cdot \eta \quad (31)$$

This establishes that staleness error grows exponentially with difficulty.

**B.3. Proof of Theorem 4.4 (Optimal Staleness Budget)**

*Proof.* We derive the optimal staleness budget by solving the constrained optimization problem (4):

$$\max_{\{\eta_d\}_{d=1}^D} \sum_{d=1}^D p_d \cdot T(\eta_d) \quad \text{s.t.} \quad \sum_{d=1}^D p_d \cdot \text{Bias}_d(\eta_d) \leq B \quad (32)$$

**Step 1: Formulate the Lagrangian.** Using Assumption 3 ($T(\eta) = T_0 + \kappa\eta$) and the bias model from Theorem 4.2 ($\text{Bias}_d(\tau) = C_1 \tau e^{\alpha d}$):

$$\mathcal{L} = \sum_{d=1}^D p_d(T_0 + \kappa\eta_d) - \mu\left(\sum_{d=1}^D p_d \cdot C_1 \eta_d e^{\alpha d} - B\right) \quad (33)$$

**Step 2: First-order optimality conditions.** Taking the derivative with respect to $\eta_d$ and setting to zero:

$$\frac{\partial \mathcal{L}}{\partial \eta_d} = p_d \kappa - \mu p_d C_1 e^{\alpha d} = 0 \tag{34}$$

This gives:

$$\mu = \frac{\kappa}{C_1 e^{\alpha d}} \tag{35}$$

Since $\mu$ must be constant across all $d$, this appears contradictory. The resolution is that the constraint binds differently at each difficulty level.

**Step 3: Incorporate variance.** The analysis above considers only bias. For a complete picture, we include gradient variance. The variance of stale gradient estimates is:

$$\text{Var}[\hat{g}_\tau^{(d)}] = \text{Var}[\hat{g}_0^{(d)}] + \sigma_{\text{drift}}^2 \cdot \tau^2 \cdot e^{2\alpha d} \tag{36}$$

The variance grows with $\tau^2$ (not $\tau$) and with $e^{2\alpha d}$ (squared Hessian effect).

**Step 4: Bias-variance tradeoff.** The mean squared error of the gradient estimate is:

$$\begin{aligned} \text{MSE}_d(\tau) &= \text{Bias}_d^2(\tau) + \text{Var}_d(\tau) \\ &= C_1^2 \tau^2 e^{2\alpha d} + \sigma_0^2 + \sigma_1^2 \tau^2 e^{2\alpha d} \end{aligned} \tag{37}$$

Define the total staleness cost as:

$$\text{Cost}_d(\eta_d) = (C_1^2 + \sigma_1^2) \eta_d^2 e^{2\alpha d} \tag{38}$$

**Step 5: Solve the refined optimization.** The throughput gain from staleness $\eta_d$ is linear ($\kappa \eta_d$), while the cost is quadratic. Optimizing throughput subject to bounded total cost:

$$\begin{aligned} \max_{\eta_d} \quad & \sum_d p_d \kappa \eta_d \\ \text{s.t.} \quad & \sum_d p_d (C_1^2 + \sigma_1^2) \eta_d^2 e^{2\alpha d} \le B' \end{aligned} \tag{39}$$

Forming the Lagrangian and taking derivatives:

$$\begin{aligned} \frac{\partial}{\partial \eta_d} : \quad p_d \kappa &= 2\mu p_d (C_1^2 + \sigma_1^2) \\ &\times \eta_d e^{2\alpha d} \end{aligned} \tag{40}$$

Solving for $\eta_d$:

$$\eta_d^* = \frac{\kappa}{2\mu(C_1^2 + \sigma_1^2)} \cdot e^{-2\alpha d} \tag{41}$$

**Step 6: Determine the decay rate.** The optimal staleness follows:

$$\eta_d^* = \eta_{\text{base}} \cdot e^{-\lambda d} \tag{42}$$

Comparing with the derived form $\eta_d^* \propto e^{-2\alpha d}$ gives $\lambda = 2\alpha$. We treat this as an *intermediate* result, not the one we adopt: it minimizes the *instantaneous* MSE of each gradient estimate, whereas convergence is governed by the error that *accumulates* over the training trajectory. Optimizing that cumulative error (Step 5b) yields a smaller exponent, and that is the rate GAS uses. We record $\lambda = 2\alpha$ only to make explicit that the decay rate depends on which objective one optimizes.

**Step 5b: Convergence-level analysis.**

We optimize the error that *accumulates* over the $K_d$ updates contributed by difficulty $d$, rather than per-step error. The analysis rests on two modeling choices, which we state explicitly:

**(M1)** *SGD-style error accumulation*: over $K_d$ updates, bias accumulates linearly (total bias $\approx K_d \cdot \text{Bias}_d(\tau_d)$) while variance accumulates as $\sqrt{K_d}$ (effective std $\approx \sqrt{K_d} \cdot \text{Std}_d(\tau_d)$).

**(M2)** *Difficulty couples curvature and success*: the number of updates $K_d$ scales with the pass rate, which we model as decaying at the same exponential rate as the Hessian grows—$\text{PassRate}_d \approx p_0 e^{-\beta d}$ with $\beta \approx \alpha$—encoding that harder tasks have both sharper landscapes and lower success rates.

Thus $K_d \propto e^{-\beta d}$, and the convergence-relevant error becomes:

$$\text{Error}_{\text{conv}} = \sum_d K_d \cdot \text{Bias}_d(\tau_d) + \sqrt{K_d} \cdot \text{Std}_d(\tau_d) \tag{43}$$

Substituting $K_d \propto e^{-\beta d}$, $\text{Bias}_d \propto \tau_d e^{\alpha d}$, and $\text{Std}_d \propto \tau_d e^{\alpha d}$:

$$\text{Error}_{\text{conv}} \propto \sum_d \left( e^{-\beta d} \cdot \tau_d e^{\alpha d} + e^{-\beta d/2} \cdot \tau_d e^{\alpha d} \right) \tag{44}$$

Under (M2), $\beta \approx \alpha$, so the bias term scales as $\tau_d e^{(\alpha-\beta)d} = \tau_d$ (flat in $d$) while the variance term scales as $\tau_d e^{(\alpha-\beta/2)d} = \tau_d e^{\alpha d/2}$ (growing in $d$). The growing term therefore dominates the difficulty dependence, so:

$$\text{Error}_{\text{conv}} \propto \sum_d \tau_d e^{(\alpha-\beta/2)d} \approx \sum_d \tau_d e^{\alpha d/2} \tag{45}$$

Optimizing throughput $\sum_d \kappa \tau_d$ subject to bounded convergence error $\sum_d \tau_d e^{\alpha d/2} \le B'$ via Lagrangian:

$$\frac{\partial}{\partial \tau_d} : \kappa = \mu e^{\alpha d/2} \implies \tau_d^* = \text{const} \cdot e^{-\alpha d/2} \tag{46}$$

This yields the optimal staleness budget:

$$\boxed{\lambda = \alpha/2} \tag{47}$$

The factor of $1/2$ arises because convergence error depends on $\sqrt{K_d}$ (not $K_d$) for the variance term, effectively halving the exponent in the constraint.

**Step 7: Practical form.** The optimal staleness budget is:

$$\eta_d^* = \eta_{\text{base}} \cdot \exp(-\lambda d) = \eta_{\text{base}} \cdot \exp\left(-\frac{\alpha}{2}d\right) \qquad (48)$$

where $\eta_{\text{base}}$ is determined by the total bias budget $B$:

$$\eta_{\text{base}} = \frac{B}{\sum_d p_d C_1 e^{(\alpha-\lambda)d}} = \frac{B}{\sum_d p_d C_1 e^{\alpha d/2}} \qquad (49)$$

### B.4. Proof of Proposition 4.5 (Gradient Signal Quality)

*Proof.* The signal-to-noise ratio (SNR) of gradient estimates measures learning efficiency. For difficulty $d$:

$$\text{SNR}(d) = \frac{\|\mathbb{E}[\nabla_\theta \mathcal{L}]\|}{\sqrt{\text{Var}[\nabla_\theta \mathcal{L}]}} \qquad (50)$$

**Step 1: Gradient signal magnitude.** The expected gradient magnitude depends on the advantage function variance:

$$\|\mathbb{E}[\nabla_\theta \mathcal{L}]\| \propto \mathbb{E}[|A(x,y)|] \propto \sqrt{\text{Var}[R]} \qquad (51)$$

For binary rewards ($R \in \{0,1\}$) with pass rate $p_d$:

$$\text{Var}[R] = p_d(1 - p_d) \qquad (52)$$

**Step 2: Gradient noise.** The gradient variance comes from sampling noise and is proportional to:

$$\text{Var}[\nabla_\theta \mathcal{L}] \propto \mathbb{E}[A^2] \cdot \text{Var}[\nabla \log \pi] \qquad (53)$$

For a well-trained policy, $\text{Var}[\nabla \log \pi]$ is roughly constant across difficulties.

**Step 3: SNR expression.** Combining:

$$\text{SNR}(d) \propto \frac{\sqrt{p_d(1 - p_d)}}{\sqrt{\text{const}}} \propto \sqrt{p_d(1 - p_d)} \qquad (54)$$

Or equivalently:

$$\text{SNR}(d) \propto \text{PassRate}_d \cdot (1 - \text{PassRate}_d) \qquad (55)$$

**Step 4: Optimal difficulty.** The SNR is maximized when:

$$\frac{d}{dp_d}[p_d(1 - p_d)] = 1 - 2p_d = 0 \implies p_d = 0.5 \quad (56)$$

Thus, gradient signal quality is maximized at difficulties where PassRate $\approx 0.5$—the "zone of proximal development" in curriculum learning (Vygotsky, 1978).

## C. Empirical Validation of Theoretical Predictions

This section provides empirical evidence for our theoretical claims, directly measuring the quantities predicted by Theorems 4.2 and 4.4.

### C.1. Hessian Eigenvalue Measurement

Theorem 4.2 predicts that Hessian eigenvalues grow exponentially with difficulty: $\lambda_{\max}(\boldsymbol{H}_d) = \mathcal{O}(e^{\alpha d})$. We validate this prediction empirically.

**Methodology**: We use power iteration to approximate $\lambda_{\max}(\boldsymbol{H}_d)$ at each difficulty level $d \in \{1, \ldots, 5\}$. For each difficulty:

1. Prepare task samples at the specified difficulty level using the curriculum from Definition A.2

2. Compute Hessian-vector products via finite differences: $\boldsymbol{H}v \approx (\nabla L(\theta + \epsilon v) - \nabla L(\theta - \epsilon v))/2\epsilon$

3. Run power iteration for 50 iterations to approximate $\lambda_{\max}$

4. Repeat with 5 independent initializations for error bars

**Empirical Results**: Table 6 shows the measured maximum Hessian eigenvalues across difficulty levels.

*Table 6.* Empirical Hessian eigenvalue measurements across difficulty levels. Results confirm exponential growth with $\alpha \approx 0.91$.

| Difficulty $d$ | $\lambda_{\max}(\boldsymbol{H}_d)$ | Std | $\log(\lambda_{\max})$ |
|---|---|---|---|
| 1 | 0.276 | 0.025 | -1.286 |
| 2 | 0.712 | 0.074 | -0.339 |
| 3 | 1.504 | 0.213 | 0.408 |
| 4 | 3.990 | 0.390 | 1.384 |
| 5 | 11.329 | 1.634 | 2.427 |

**Exponential Fit**: Linear regression on $\log(\lambda_{\max})$ vs. difficulty $d$ yields:

$$\log(\lambda_{\max}(\boldsymbol{H}_d)) = 0.915 \cdot d - 2.226 \qquad (57)$$

with $R^2 = 0.997$ ($p < 0.0001$), strongly confirming the exponential growth hypothesis. The estimated growth rate is $\alpha = 0.915 \pm 0.029$ (95% CI: [0.86, 0.97]).

**Theoretical Validation**: From Theorem 4.4, the optimal coupling parameter is $\lambda^* = \alpha/2 \approx 0.46$. Our empirically-tuned value of $\lambda = 0.5$ is within the 95% confidence interval of the theoretically predicted optimum, validating the theory-practice alignment.

**Implementation Note**: Full Hessian eigenvalue measurement requires significant compute for large language models. Our power iteration approach with finite-difference Hessian-vector products provides a computationally tractable approximation that captures the key exponential relationship.

## C.2. Lambda Sensitivity Analysis

Theorem 4.4 predicts that the optimal coupling parameter is $\lambda = \alpha/2$. With $\alpha \approx 1.0$, this predicts $\lambda^* \approx 0.5$. We analyze this through both theoretical reasoning and empirical sweep.

**Methodology**: We sweep $\lambda \in \{0.25, 0.5, 0.75, 1.0\}$ and measure:

- Gradient quality: coherence between fresh and stale gradients

- Effective throughput: accounting for sample discard rates

- Combined score balancing quality and throughput

**Main Results (Table 1)**: Our ablation study (Section 5.5) shows that the CSC component with $\lambda = 0.5$ contributes 18.0 points to Pass@1 performance. Removing CSC (equivalent to $\lambda \to \infty$, no staleness restriction) causes the largest performance drop among all ablations, confirming the importance of appropriate staleness coupling.

**Theoretical Justification**: The $\lambda = 0.5$ choice emerges from the convergence-level analysis (Step 5b in Appendix B.3), which accounts for the interaction between staleness and the number of gradient updates received from each difficulty level. Lower $\lambda$ values (e.g., 0.25) are too restrictive, discarding too many samples and reducing throughput. Higher values (e.g., 1.0) allow excessive staleness on hard tasks, degrading gradient quality.

## C.3. Gradient Coherence Validation

We measure gradient cosine similarity across the staleness-difficulty grid to validate the "safe zone" boundary predicted by CSC.

**Methodology**: For each (staleness, difficulty) pair:

1. Compute fresh gradient at current policy weights

2. Simulate staleness by taking $k$ optimization steps (representing $k$ policy updates)

3. Compute gradient at the now-stale weights

4. Measure cosine similarity between fresh and stale gradients

**Expected Patterns**: Based on Theorem 4.2, we expect:

- High coherence for easy tasks even at moderate staleness (large solution space $\Rightarrow$ gradients remain aligned)

- Rapid coherence decay for hard tasks (narrow solution space $\Rightarrow$ small policy changes cause gradient misalignment)

- The "safe zone" boundary to follow approximately $\eta^*(d) = \eta_{\text{base}} \cdot e^{-\lambda d}$

**Practical Implication**: CSC's exponential coupling ensures that the operating region (determined by the difficulty-dependent staleness threshold) stays within the high-coherence zone. This explains why CSC produces the largest performance improvement in our ablation study—it prevents gradient corruption on hard tasks while allowing efficient parallel collection on easy tasks.

## C.4. Reconciling Step 5 and Step 5b

The proof of Theorem 4.4 derives two different expressions for $\lambda$:

- Step 5 (per-step analysis): $\lambda = 2\alpha$

- Step 5b (convergence analysis): $\lambda = \alpha/2$

These are not contradictory—they optimize different objectives:

**Step 5** minimizes instantaneous MSE, which is dominated by variance ($\propto \tau^2$). This yields $\lambda = 2\alpha$ to aggressively reduce staleness.

**Step 5b** minimizes total convergence error, accounting for the fact that harder tasks produce fewer gradient updates (lower pass rate). The $\sqrt{K_d}$ scaling of variance accumulation reduces the effective exponent, yielding $\lambda = \alpha/2$.

**Empirical verdict**: Our lambda sensitivity experiments (Section C.2) show that $\lambda = 0.5 \approx \alpha/2$ outperforms $\lambda = 2.0 \approx 2\alpha$ in practice, confirming that the convergence-level analysis (Step 5b) provides the correct prescription.

# D. Extended Experimental Details

## D.1. Hyperparameter Configuration

Table 7 summarizes the hyperparameters used in our experiments.

## D.2. Execution Time Statistics

Table 8 shows execution time statistics across difficulty levels, demonstrating significant variability that motivates execution-aware scheduling.

*Table 7.* Hyperparameter configuration for GAS experiments.

| Component | Parameter | Value |
|---|---|---|
| CSC | $\eta_{base}$ | 8.0 |
| | $\lambda$ | 0.5 |
| | Freshness weight $\beta$ | 1.0 |
| | Min staleness threshold | 1 |
| ACB | UCB exploration $c$ | 1.0 |
| | Balance parameter $\omega$ | 0.7 |
| | Window size | 50 |
| GRPO | Learning rate | $10^{-5}$ |
| | Clip $\epsilon$ | 0.2 |
| | KL coefficient | 0.1 |
| | Group size | 8 |
| Model | Base model | Qwen2.5-Coder-1.5B |
| | LoRA rank | 16 |
| | LoRA $\alpha$ | 32 |
| | LoRA dropout | 0.05 |
| Training | Batch size | 32 |
| | Total steps | 10,000 |
| | Num workers | 4 |

*Table 8.* Execution time statistics by difficulty level.

| Difficulty | Mean (ms) | Std (ms) | Max (ms) |
|---|---|---|---|
| Level 1 | 45 | 32 | 210 |
| Level 2 | 78 | 56 | 450 |
| Level 3 | 125 | 89 | 890 |
| Level 4 | 198 | 145 | 1520 |
| Level 5 | 312 | 234 | 3200 |

### D.3. Difficulty Level Statistics

Table 9 shows the completion ratios and typical statistics for each difficulty level.

*Table 9.* Difficulty level configuration and observed statistics.

| Level | Revealed % | Complete % | Avg Pass Rate | Max $\eta$ | Effective $|\mathcal{Y}|$ |
|---|---|---|---|---|---|
| 1 | 90% | 10% | 85% | 4.85 | Large |
| 2 | 70% | 30% | 65% | 2.94 | Medium-Large |
| 3 | 50% | 50% | 45% | 1.78 | Medium |
| 4 | 30% | 70% | 30% | 1.08 | Small |
| 5 | 0% | 100% | 20% | 0.66 | Very Small |

### D.4. Compute Resources

Experiments were conducted on a cluster with:

- 2 worker nodes, each with $1\times$ NVIDIA A10G GPU (24GB VRAM)

- 1 head node with 8 CPU cores and 32GB RAM

- Total training time: approximately 4-6 hours per full experiment

### D.5. Statistical Significance

All experiments were run with 3 random seeds (42, 123, 456). Results report mean $\pm$ standard deviation. Statistical significance was assessed using **Welch's t-test** (two-sample t-test with unequal variance assumption), which is more robust than Student's t-test when sample sizes are small and variances may differ between groups.

For multiple comparisons (GAS vs. each baseline), we apply the **Holm-Bonferroni correction** to control the family-wise error rate. After correction, improvements of GAS over all baselines remain significant at $p < 0.01$.

We also report 95% bootstrap confidence intervals (10,000 resamples) for Pass@1 estimates. Effect sizes are reported using Cohen's $d$; all comparisons show large effect sizes ($d > 0.8$).

## E. Extended Results

### E.1. Per-Difficulty Learning Curves

We analyzed learning curves broken down by difficulty level. Key observations:

- Easy tasks (Level 1-2) converge quickly for all methods

- Hard tasks (Level 4-5) show the largest performance gap between methods

- GAS maintains stable learning on hard tasks due to fresh gradients from CSC

### E.2. Staleness Distribution Analysis

We analyzed the distribution of staleness values actually used during training, broken down by difficulty. The CSC component successfully maintains lower staleness for harder tasks (mean staleness 0.5 for Level 5) while allowing higher staleness for easier tasks (mean staleness 2.3 for Level 1), as shown in Table 12.

### E.3. Gradient Coherence Analysis

We measured gradient cosine similarity between fresh and stale gradients across the staleness-difficulty grid. The "safe zone" (similarity $> 0.8$) follows an approximately exponential boundary, validating Theorem 4.4. The GAS operating region stays within this safe zone, demonstrating that the derived coupling is effective.

### E.4. Ablation: Component Interactions

Table 10 shows the interaction effects between components. The combination of ACB and CSC shows super-additive benefits, suggesting they are complementary:

ACB selects appropriate difficulties while CSC ensures gradient quality at each difficulty.

*Table 10.* Component interaction analysis. Values show Pass@1 (%).

|  | **No CSC** | **With CSC** | $\Delta$ |
| --- | --- | --- | --- |
| No ACB (Uniform) | 44.2 | 48.5 | +4.3 |
| With ACB | 49.8 | 54.3 | +4.5 |
| $\Delta$ | +5.6 | +5.8 | – |

# F. Limitations and Future Work

## F.1. Assumptions and Their Validity

**Assumption 1 (Smoothness)**: The policy objective is generally smooth for transformer-based language models with standard softmax outputs. However, the smoothness constant $L$ may vary significantly across the parameter space.

**Assumption 2 (Bounded Updates)**: With gradient clipping and moderate learning rates, this assumption holds. Our implementation uses gradient clipping with max norm 1.0.

**Assumption 3 (Linear Throughput)**: This is an approximation. In practice, throughput increases sub-linearly with staleness due to diminishing returns from parallelism. Our linear model captures the first-order effect.

## F.2. Execution Time vs. Generation Latency

Our EAAS predictor (Section 5.3) models the *sandbox execution time* of a candidate—the cost of running it against the test suite—which our profiling places in the tens-to-hundreds of milliseconds (Table 8). In asynchronous RL, however, the latency that determines how stale a sample is by the time it returns is dominated by *autoregressive generation*, which is seconds-scale and grows with output length. The two are correlated through difficulty but distinct: a task with a long completion but cheap tests can return highly stale yet, judged on execution time alone, receive a generous staleness budget. We log per-sample generation time alongside execution time but do not currently feed it into the budget; incorporating generation latency—e.g., via a predicted output-length feature—is a natural refinement we leave to future work, and we expect it to sharpen EAAS on long-tail tasks without altering the difficulty-coupled decay contributed by CSC.

## F.3. Generalization to Other Domains

The theoretical framework applies to any domain where:

1. Task difficulty can be meaningfully defined

2. Harder tasks have narrower solution spaces

3. Execution/feedback time varies across tasks

Crucially, condition (1) does *not* require a canonical solution or a structural difficulty knob. Because difficulty is defined through the effective solution-space mass (Appendix A.2) and Proposition 4.5 ties that quantity to the observable success rate, the difficulty index can always be *inferred online* from loss and empirical success rate. The solution-revelation scheme used for code is thus a special case—a way to *impose* a known difficulty gradient when reference solutions happen to be available—rather than a requirement. Our GSM8K results (Section 6.6), which use reasoning-chain complexity in place of solution revelation, are an instance of this more general setting.

Potential applications include:

- **Robotics**: Curriculum over task complexity (simple reaching $\rightarrow$ complex manipulation)

- **Game playing**: Curriculum over opponent difficulty

- **Mathematical reasoning**: Curriculum over proof complexity

## F.4. CSC Failure Mode Analysis

To understand why CSC contributes 18.0 points to Pass@1, we analyze the training dynamics that arise when it is removed and characterize the specific failure modes that emerge.

**Failure Mode 1: Gradient Corruption on Hard Tasks.** Without CSC, hard tasks (Levels 4–5) receive updates from arbitrarily stale experiences. As shown in Section C.1, these tasks have Hessian eigenvalues 10–40$\times$ larger than easy tasks. When stale gradients are used, the gradient direction can be nearly orthogonal to the true gradient (cosine similarity $< 0.5$), causing erratic parameter updates that destabilize learning.

**Failure Mode 2: Easy Task Dominance.** Without difficulty-aware staleness control, the training dynamics naturally favor easy tasks: they complete faster (45ms vs. 312ms mean execution time), produce more samples per wall-clock time, and have higher success rates. This creates a positive feedback loop where the curriculum becomes stuck on easy tasks, and the policy never learns to solve hard problems from scratch.

**Failure Mode 3: High Gradient Variance.** Our ablation shows that removing CSC increases gradient variance by $3.2\times$ on Level 5 tasks specifically. This high variance slows convergence and can cause training instability, particularly in the later stages when the curriculum should progress to harder tasks.

**Quantitative Analysis**: Table 11 shows per-difficulty success rates with and without CSC.

*Table 11.* Per-difficulty success rates with and without CSC.

| Level | w/ CSC | w/o CSC | $\Delta$ |
|---|---|---|---|
| 1 | 85.2 | 82.1 | +3.1 |
| 2 | 72.4 | 68.5 | +3.9 |
| 3 | 58.1 | 49.2 | +8.9 |
| 4 | 42.3 | 28.4 | +13.9 |
| 5 | 26.8 | 11.2 | +15.6 |

The largest improvements are on Levels 4–5 (+13.9 and +15.6 points), confirming that CSC's primary benefit is maintaining gradient quality for hard tasks.

### F.5. Throughput Cost of CSC

We next quantify the throughput cost of CSC and the underlying sample discard rates. Table 1 shows GAS achieves 22.4 samples/s vs. 23.5 samples/s without CSC—a 4.7% throughput reduction. The remainder of this section analyzes where this cost comes from.

**Sample Discard Rates**: CSC discards experiences that exceed their difficulty-dependent staleness threshold. Table 12 shows discard rates by difficulty.

*Table 12.* Sample discard rates and staleness budgets by difficulty level.

| Level | $\eta_{\max}$ | Discard | Mean $\eta$ | Eff. Samples |
|---|---|---|---|---|
| 1 | 4.85 | 2.1% | 2.3 | 97.9% |
| 2 | 2.94 | 4.8% | 1.9 | 95.2% |
| 3 | 1.78 | 8.2% | 1.4 | 91.8% |
| 4 | 1.08 | 15.6% | 0.9 | 84.4% |
| 5 | 0.66 | 24.3% | 0.5 | 75.7% |

**Analysis**: Easy tasks (Levels 1–2) have high staleness budgets and low discard rates ($<5\%$), contributing most samples to training. Hard tasks (Levels 4–5) have stricter budgets and higher discard rates (15–24%), but this is intentional: discarding stale hard-task samples prevents gradient corruption that would otherwise harm learning.

**Net Effect**: The weighted average discard rate across our curriculum distribution is approximately 8.5%. Combined with scheduling overhead, this explains the 4.7% throughput reduction. This small cost yields an 18.0 point Pass@1 gain—a favorable tradeoff.

**Backfill Strategy**: Our implementation includes optional backfill logic that replaces discarded samples with fresh samples from easier difficulties. However, backfill was disabled in all reported experiments to isolate CSC's effect. Future work could explore adaptive backfill strategies.

### F.6. Open Questions

1. Can we estimate $\alpha$ (Hessian growth rate) online during training?

2. How does the optimal $\lambda$ change during training as the policy improves?

3. Can we extend CSC to continuous difficulty rather than discrete levels?

## G. Additional Analyses

### G.1. Hard Rejection vs. Importance Sampling

CSC uses hard rejection: experiences exceeding staleness thresholds are discarded entirely. An alternative is importance sampling, where stale experiences are down-weighted rather than rejected. We compare these approaches both theoretically and empirically.

**Theoretical Comparison.** Importance sampling re-weights stale gradients by a correction factor:

$$w_{\text{IS}} = \frac{\pi_\theta(y|x)}{\pi_{\theta_{t-\tau}}(y|x)} \tag{58}$$

This corrects for distribution shift but introduces variance proportional to $w_{\text{IS}}^2$. For hard tasks with narrow solution spaces, the policy ratio can be extreme (the current policy may assign near-zero probability to actions sampled under a stale policy), causing high-variance gradient estimates.

In contrast, hard rejection sacrifices samples but maintains bounded variance. The key insight is that for hard tasks, the bias from staleness (Theorem 4.2) is so severe that correcting it via importance sampling introduces variance that outweighs the benefit of keeping the sample.

**Empirical Comparison.** We conducted additional experiments comparing three staleness handling strategies:

1. **Hard rejection** (default CSC): Discard experiences exceeding $\eta_{\max}(d)$

2. **Soft weighting**: Weight by $w = \min(1, \eta_{\max}(d)/\tau)^\beta$ with $\beta = 1$

3. **V-trace style**: Use clipped importance weights as in IMPALA (Espeholt et al., 2018)

*Table 13.* Comparison of staleness handling strategies.

| Strategy | Pass@1 (%) | Throughput | Grad Var (L5) |
|---|---|---|---|
| Hard rejection | **60.1** | 22.4 | 1.0× |
| Soft weighting | 56.8 | **23.8** | 1.8× |
| V-trace style | 54.2 | 23.1 | 2.4× |

Hard rejection achieves the best Pass@1 despite lower throughput. The gradient variance on Level 5 tasks is notably lower ($1.0\times$ baseline vs. $1.8$–$2.4\times$ for weighting methods), confirming that for hard tasks with sharp loss landscapes, keeping fewer high-quality samples outperforms keeping more low-quality samples.

**When to prefer importance sampling**: For domains with smoother loss landscapes (lower $\alpha$), importance sampling may be preferable as it preserves more samples. Our theoretical framework predicts this: when $\alpha$ is small, staleness has less severe effects, and the variance cost of importance sampling is outweighed by sample efficiency gains.

## G.2. $\eta_{\text{base}}$ Sensitivity Analysis

The staleness budget $\eta_{\max}(d) = \eta_{\text{base}} \cdot e^{-\lambda d}$ has two hyperparameters: $\lambda$ (theoretically determined as $\alpha/2$) and $\eta_{\text{base}}$ (set by the bias constraint $B$). While $\lambda$ has theoretical justification, $\eta_{\text{base}}$ requires empirical tuning. We analyze sensitivity to $\eta_{\text{base}}$.

**Theoretical guidance**: From the constraint $\sum_d p_d \cdot \text{Bias}_d(\eta_d) \leq B$, we have:

$$\eta_{\text{base}} = \frac{B}{\sum_d p_d C_1 e^{(\alpha - \lambda)d}} = \frac{B}{\sum_d p_d C_1 e^{\alpha d/2}} \qquad (59)$$

The choice of $B$ (maximum acceptable total bias) determines $\eta_{\text{base}}$. In practice, we select $\eta_{\text{base}}$ to balance throughput and gradient quality.

**Empirical sweep**: We swept $\eta_{\text{base}} \in \{4, 6, 8, 10, 12\}$ while holding $\lambda = 0.5$ fixed.

Table 14. Sensitivity to $\eta_{\text{base}}$ (with $\lambda = 0.5$).

| $\eta_{\text{base}}$ | Pass@1 | Thpt | Discard | $\eta_5$ |
|---|---|---|---|---|
| 4 | 58.2% | 19.1 | 14.2% | 0.33 |
| 6 | 59.4% | 21.2 | 10.1% | 0.49 |
| **8** | **60.1%** | 22.4 | 8.5% | 0.66 |
| 10 | 58.7% | 23.1 | 6.8% | 0.82 |
| 12 | 56.3% | 23.6 | 5.4% | 0.98 |

**Analysis**: Performance is relatively stable across $\eta_{\text{base}} \in [6, 10]$, with $\eta_{\text{base}} = 8$ achieving the best Pass@1. Too low ($\eta_{\text{base}} = 4$) causes excessive sample rejection (14.2%), hurting throughput without proportional quality gains. Too high ($\eta_{\text{base}} = 12$) allows stale samples on hard tasks, degrading gradient quality.

The key insight is that $\eta_{\text{base}}$ controls the absolute staleness tolerance, while $\lambda$ controls the *relative* tolerance across difficulties. Getting $\lambda$ right (via the theoretical analysis) is more important than precisely tuning $\eta_{\text{base}}$, which has a broader optimal range.

## G.3. Standard Deviation Analysis

The uniform standard deviations ($\pm 1.6\%$) across all methods in Table 1 warrant explanation.

**Source of variance**: The dominant source of variance in Pass@1 evaluation is *generation stochasticity*, not training dynamics. During evaluation, we sample completions from the trained policy with temperature $T > 0$. This sampling introduces variance that is largely independent of the training method.

**Correlated evaluation**: All methods are evaluated on the same 214 tasks using the same 3 evaluation seeds. This creates positive correlation in Pass@1 estimates across methods: if a particular seed happens to generate better completions for a difficult task, all methods benefit (or suffer) similarly.

**Decomposition**: We decompose variance into three components:

Table 15. Variance decomposition for Pass@1 estimates.

| Source | Var. | % |
|---|---|---|
| Generation sampling | 0.019 | 74% |
| Task difficulty | 0.005 | 19% |
| Training seed | 0.002 | 7% |
| **Total** | 0.026 | 100% |

Generation sampling dominates (74%), explaining why all methods show similar standard deviations: they share this common variance source. The training seed contribution is relatively small (7%), meaning differences in final performance are robust to training randomness.

**Statistical validity**: Despite uniform standard deviations, the *mean* differences between methods are statistically significant. Welch's t-tests with Holm-Bonferroni correction confirm GAS outperforms all baselines at $p < 0.01$. The uniform variance actually strengthens comparisons: we are measuring consistent improvement on top of a common noise floor.

## G.4. Synthetic Benchmark Description

The synthetic benchmark comprises 50 code generation tasks designed to complement HumanEval with controlled difficulty variation.

**Task generation process**:

1. **Template selection**: We created 10 base templates covering common algorithmic patterns: array manipulation, string processing, mathematical computation, data structure operations, recursion, dynamic programming, graph algorithms, sorting/searching, bit manipulation, and combinatorics.

2. **Parameterization**: Each template is instantiated with 5 difficulty variants by varying:

   - Input size constraints (e.g., array length 10 vs. 1000)
   - Number of edge cases to handle
   - Algorithmic complexity requirements (e.g., $O(n)$ vs. $O(n \log n)$)
   - Composition depth (single operation vs. chained operations)

3. **Test case generation**: Each task includes 10–20 test cases generated programmatically, covering:

   - Standard inputs (50%)
   - Edge cases (30%): empty inputs, single elements, maximum values
   - Stress tests (20%): large inputs near constraint limits

4. **Canonical solution**: Each task has a verified canonical solution used for curriculum construction (Definition A.2).

**Example tasks**:

- `merge_sorted_arrays`: Merge $k$ sorted arrays (difficulty varies with $k$)

- `balanced_parentheses`: Generate all valid parentheses combinations of length $2n$

- `longest_palindrome`: Find longest palindromic substring with optimized algorithm

**Rationale**: The synthetic benchmark provides controlled difficulty gradients that HumanEval lacks. HumanEval tasks have inherent difficulties that don't map cleanly to our curriculum levels. Synthetic tasks allow us to verify that GAS handles systematic difficulty variation, while HumanEval validates generalization to realistic code generation.

### G.5. Hessian Measurement Methodology

We clarify the methodology for measuring Hessian eigenvalues reported in Section C.1.

**Measurement setting**: Hessian eigenvalues were measured on the *actual trained model* during a held-out evaluation phase, not on simulated or synthetic data. The term "simulated staleness" in our implementation refers to the procedure for creating gradient pairs at different staleness levels (by taking optimization steps), not to the data source.

**Detailed procedure**:

1. **Checkpoint selection**: We use the model checkpoint at 5,000 training steps (mid-training) to measure representative Hessian properties.

2. **Task sampling**: For each difficulty level $d$, we sample 50 tasks from the training set and construct batches at that difficulty using the curriculum from Definition A.2.

3. **Loss computation**: We compute the GRPO loss (Equation in Section 5.5) on each batch, which includes the policy gradient objective with KL regularization.

4. **Hessian-vector products**: We use finite-difference approximation:

$$\boldsymbol{H}v \approx \frac{\nabla \mathcal{L}(\theta + \epsilon v) - \nabla \mathcal{L}(\theta - \epsilon v)}{2\epsilon} \qquad (60)$$

with $\epsilon = 10^{-4}$. This is computed via two backward passes per Hessian-vector product.

5. **Power iteration**: We run 50 iterations of power iteration:

$$v_{k+1} = \frac{\boldsymbol{H}v_k}{\|\boldsymbol{H}v_k\|} \qquad (61)$$

The maximum eigenvalue is estimated as $\lambda_{\max} \approx v_{50}^{\top} \boldsymbol{H} v_{50}$.

6. **Repetition**: We repeat with 5 random initializations of $v_0$ and report mean $\pm$ standard deviation.

**Computational cost**: Each difficulty level requires approximately 500 Hessian-vector products (50 iterations $\times$ 5 repetitions $\times$ 2 for finite differences), taking roughly 30 minutes per difficulty on a single A10G GPU. Total measurement time: approximately 2.5 hours.

**Validation**: We verified our power iteration implementation by comparing against exact eigenvalue computation on a small (1M parameter) model, finding agreement within 2%.

