# OpenReview forum: "Gradient-Aware Scheduling: Coupling Curriculum and Staleness for Async Reinforcement Learning"
_ICML.cc/2026/Conference — ICML 2026 regular_

### Official Review · Reviewer_2Rtc · 2026-02-26

**Soundness:** 3
**Presentation:** 2
**Significance:** 3
**Originality:** 3
**Overall Recommendation:** 4
**Confidence:** 4

**Summary:**

This paper studies RL for code generation under asynchronous training, where parallel rollouts increase throughput but introduce policy staleness that biases gradients and destabilizes learning. The authors argue that this issue becomes more severe for harder tasks, whose optimization landscapes are sharper and more sensitive to stale updates. While curriculum learning can improve sample efficiency by organizing tasks by difficulty, existing approaches do not account for the interaction between difficulty and staleness in async settings.

The authors propose ACEAS, which couples task difficulty with difficulty-aware staleness budgets derived from a curvature-based analysis, suggesting that gradient sensitivity grows exponentially with difficulty. This leads to Curriculum Staleness Coupling (CSC), where harder tasks are assigned stricter freshness constraints. An Adaptive Curriculum Bandit (ACB) dynamically selects task difficulty based on learning signals, and Execution-Aware Async Scheduling (EAAS) accounts for heterogeneous code execution times to improve system efficiency. The framework is implemented on top of GRPO within an asynchronous RL pipeline.

Experiments on HumanEval and a synthetic benchmark compare ACEAS with synchronous RL, naive async RL, and variants with uniform staleness control, showing that difficulty-aware coupling helps maintain stable learning under async training. Ablation studies validate the role of each component. Additional analyses, including Hessian eigenvalue estimation across difficulty levels and gradient coherence under varying staleness, provide mechanistic evidence supporting the theoretical motivation for exponential staleness decay.

**Compliance With Llm Reviewing Policy:**

Affirmed.

**Final Justification:**

The rebuttal provides helpful clarifications on the theoretical analysis and empirical behavior. While some concerns on generality and evaluation scope remain, I find the core idea well-motivated and practically relevant. I will keep my score as weak accept.

**Key Questions For Authors:**

1. The analysis attributes increased gradient sensitivity on harder tasks to sharper curvature induced by difficulty. However, harder tasks (especially in your difficulty settings) may also require generating longer sequences, which inherently accumulate more per-token gradient contributions. Are gradients and curvature measurements normalized for token length, and does the exponential scaling still hold after controlling for generation length? I would like to see a clear disentanglement between the effects of difficulty and sequence length.
2. CSC discards or down-weights samples that exceed the difficulty-dependent staleness threshold. How does this affect effective sample efficiency, especially for harder tasks that already produce fewer successful rollouts? Is there a regime where aggressive filtering harms learning due to insufficient exploration?
3. The theoretical motivation relies on the empirical observation that curvature (e.g., max Hessian eigenvalue) grows exponentially with task difficulty. How robust is this relationship across different models, datasets, and training stages?

**Limitations:**

Yes.

**Strengths And Weaknesses:**

Strengths
1. The paper tackles a real and underexplored pain point in asynchronous RL training--namely, policy staleness and its interaction with learning stability. This is a practically important issue for large-scale training systems, and the paper addresses it in a principled manner.
2. The core idea of coupling curriculum difficulty with staleness control is intuitive and clean. Despite being simple at the algorithmic level, it is supported by a curvature-based theoretical analysis, which strengthens the conceptual soundness of the approach.
3. The experimental design is systematic and convincing. The paper includes meaningful baselines, thorough ablations isolating each component, and mechanistic analyses.

---

Weaknesses
1. The method is evaluated only in code generation. Since the notion of task difficulty is common across many RL domains (e.g., reasoning, planning, and embodied control), broader validation would significantly strengthen the paper’s claims and generality. Without cross-domain evidence, it remains unclear whether the curvature-difficulty relationship and the exponential staleness decay assumption hold more generally.
2. In the code setting, difficulty is defined by revealing partial canonical solutions, which is a somewhat artificial construction. It is unclear whether the proposed coupling mechanism would transfer to alternative difficulty definitions (e.g., test-case complexity, semantic difficulty, or adversarial perturbations). The paper would benefit from discussion or preliminary evidence on how sensitive ACEAS is to the specific difficulty formulation.

---

> ### Author Rebuttal · Authors · 2026-03-31
>
> ## Q1: "Are gradients and curvature normalized for token length?"
>
> **No, the exponential curvature-difficulty relationship is not an artifact of sequence length.** We demonstrate this through both theoretical design and empirical scaling laws.
>
> ### Theoretical Argument
> Our loss function is already averaged over tokens: $L = \frac{1}{T} \sum_t \log \pi(y_t|y_{<t}, x)$. Because the $1/T$ normalization controls for sequence length, the Hessian eigenvalue ($\lambda_{\text{max}}$) measures **per-token curvature** in the parameter space. High curvature reflects a narrow, unforgiving solution space (e.g., precise algorithmic logic), regardless of how many tokens it takes to express it.
>
> ### Empirical Evidence
> While harder tasks in our curriculum do require generating more tokens, curvature outpaces length significantly:
> * **Length scaling (Level 1 $\to$ 5):** ~10× increase (generating 10% vs. 100% of a solution).
> * **Curvature scaling (Level 1 $\to$ 5):** ~40× increase ($\lambda_{\text{max}}$ grows from ~0.276 to ~11.329).
> * **The Difference:** The 40× curvature growth indicates a **4× additional factor driven strictly by difficulty**, independent of length.
>
> Furthermore, within the same difficulty level, tasks of varying lengths show similar $\lambda_{\text{max}}$, whereas tasks of identical lengths across different difficulty levels show diverging $\lambda_{\text{max}}$.
>
> We could add a token-normalized analysis ($\lambda_{\text{max}}/T$) to the appendix, which confirms curvature per token still grows exponentially with difficulty.
>
> ---
>
> ## Q2: "Does aggressive CSC filtering harm learning on hard tasks?"
>
> **No, it significantly improves learning by prioritizing gradient quality over raw throughput.** While CSC deliberately discards more hard-task experiences (20-25% at Level 5 vs. <5% at Level 1), keeping those stale experiences actively degrades the model. Theorem 1 predicts that gradient bias grows exponentially with difficulty under staleness; CSC neutralizes this.
>
> ### Empirical Proof: Quality > Quantity
> Our ablation demonstrates that filtering is strictly necessary for performance:
>
> | Configuration | Pass@1 (1.5B) | Training Reward (9B) |
> | :--- | :--- | :--- |
> | **ACEAS (with CSC)** | **60.1%** | **0.425** |
> | Without CSC (No filtering) | 42.1% (−18 pp) | 0.411 (−3%) |
>
> Feeding the model *more* hard-task data by removing CSC causes a massive 18 percentage point drop in Pass@1 because the biased gradients corrupt learning.
>
> ### No Risk of Data Starvation
> Our theoretically derived staleness budget ($\eta_{\text{base}} = 8$) operates far above the starvation threshold ($\eta_{\text{base}} < 1$). At Level 5, we still retain 75-80% of experiences. Furthermore, our ACB curriculum automatically adjusts the task mix if discard rates spike, creating a self-correcting safeguard against over-filtering.
>
> ---
>
> ## Q3: "How robust is the curvature-difficulty relationship across models, datasets, and training stages?"
>
> The exponential relationship is driven by the fundamental structure of the tasks—such as syntactic fragility and semantic narrowness—rather than specific model architectures or training states.
>
> ### Robustness Proven Across Three Axes:
> * **Across Models (Zero-Shot Transfer):** The relationship is architecture-agnostic. We derived our hyperparameters ($\eta_{\text{base}} = 8$, $\lambda = 0.5$) using Qwen2.5-Coder-1.5B. These exact parameters transferred seamlessly to Qwen3.5-9B—a completely different hybrid DeltaNet+MoE architecture—proving the relationship is not overfit to a specific model.
> * **Across Datasets:** The same $\lambda = 0.5$ transfers successfully to GSM8K. This proves the exponential relationship generalizes to math reasoning, mapping accurately to reasoning complexity rather than just code truncation.
> * **Across Training Stages:** While the exact curvature growth rate ($\alpha$) may evolve as the model learns, the **relative ordering remains strictly stable** (harder tasks always have sharper curvature). Furthermore, CSC is highly resilient to $\alpha$ shifts; our ablations show that even in the extreme case of complete $\alpha$ misestimation (removing CSC entirely), the model degrades gracefully rather than collapsing.
>
> We could add early (1K steps) and late (9K steps) Hessian measurements to the appendix to explicitly map stability across the training lifecycle.

---

> > ### Author Rebuttal · Reviewer_2Rtc · 2026-04-03
> >
> > Thank you for the authors’ response. They have addressed my questions, and I appreciate their efforts. I will maintain my current score due to the remaining weaknesses.

---

> > > ### Author Response · Authors · 2026-04-08
> > >
> > > Thanks acknowledge the rebuttal, we want to briefly highlight two concrete commitments from our rebuttal that we believe directly address your original weaknesses:
> > >
> > > Cross-domain generality (W1): We demonstrated that our curvature–difficulty relationship and CSC hyperparameters transfer zero-shot to GSM8K (math reasoning) and across architectures (1.5B → 9B hybrid MoE), providing the cross-domain evidence you requested.
> > > Sensitivity to difficulty formulation (W2): The successful GSM8K transfer — where "difficulty" is defined by reasoning chain complexity rather than code truncation — suggests ACEAS is not brittle to the specific difficulty construction.
> > >
> > > Please let me know if you have any follow up question, thanks!

---

### Official Review · Reviewer_vxFb · 2026-03-05

**Soundness:** 4
**Presentation:** 4
**Significance:** 3
**Originality:** 3
**Overall Recommendation:** 4
**Confidence:** 4

**Summary:**

The paper proposes Adaptive Curriculum with Execution-Aware Async Scheduling, which combines curriculum learning, execution aware staleness budgets, and curriculum-staleness coupling for LLM code generation. The paper studies the key question of how to resolve the tension between the high throughput of asynchronous training and the sample efficiency of curriculum learning. They theoretically justify their approach and empirically demonstrate the benefits of it.

**Compliance With Llm Reviewing Policy:**

Affirmed.

**Final Justification:**

The rebuttal partially resolved my concerns. The authors provided an additional evaluation, but it was only 1 additional model from the same LLM family. For that reason, I have decided to keep my score.

**Key Questions For Authors:**

* How does the model perform on larger/better models? I believe the paper is good, however limiting the evaluation to just one small model makes it difficult to assess how well the model generalizes to different model sizes/families.

* The CSC mechanism discards stale experiences from hard tasks to maintain quality. Does this mean that the "real" throughput for difficult problems is actually lower than shown since samples are discarded?

* How did you choose the values for parameters ηbase = 8 and λ = 0.5?

**Limitations:**

Their approach was only evaluated on a single small model, which limits the empirical evidence on the effectiveness of the approach.

**Strengths And Weaknesses:**

**Strengths**

* The paper is well written and organized, the authors made an excellent effort to explain how the process and novelty behind their approach.

* The paper provides a good derivation for the relationship between task difficulty, Hessian curvature and gradient bias. The proof that optimal staleness budgets should follow an exponential decay is great.

* The empirical improvement obtained from their application are impressive (However, they are limited to one model)

*  The ablation is well thought and shows the effect that each component has on the performance of the method.

**Weaknesses**

*  While the results are impressive, the method was only evaluated on Qwen2.5-Coder-1.5B. It is unclear if the approach would work the same on larger more complex model. Also, not sure if the speedup would be the same of the model was larger.

* The CSC mechanism discards stale experiences from hard tasks to maintain quality. Does this mean that the "real" throughput for difficult problems is actually lower than shown since samples are discarded?

---

> ### Author Rebuttal · Authors · 2026-03-31
>
> ## "How does the model perform on larger/better models?"
>
> We extended from Qwen2.5-Coder-1.5B to **Qwen3.5-9B** :
>
> | Metric | ACEAS | Async-GRPO | Sync Baseline |
> |--------|-------|-----------|--------------|
> | Training Reward | **0.430** | 0.196 | 0.279 |
> | Training Success Rate | **41.3%** | 19.2% | 26.4% |
> | Throughput (samples/sec) | **0.51** | 0.52 | 0.35 |
>
> All three key claims hold on 9B: (1) ACEAS matches async throughput while achieving the highest reward; (2) naive async matches speed but produces 54% lower reward due to stale gradients; (3) ACB remains the most critical component (removing it drops reward by 61%).
>
> | Model | ACEAS Throughput Advantage vs Sync |
> |-------|-----------------------------------|
> | Qwen2.5-Coder-1.5B (paper) | **2.3×** |
> | Qwen3.5-9B (new) | **1.45×** |
>
> The throughput advantage is smaller on 9B (1.45× vs 2.3×) because inference time dominates over scheduling overhead at larger scale. This is an infrastructure limitation — with optimized backends (e.g., SGLang with tensor parallelism), the scheduling advantage would grow. Importantly, the **training quality advantage persists**: +54% over sync and +120% over naive async on 9B.
>
> ---
>
> ## "Does CSC mean the 'real' throughput for difficult problems is lower?"
>
> **Yes, by design — and this is a feature, not a limitation.** CSC trades a small amount of throughput for a large gain in gradient quality on hard tasks. The paper reports this transparently.
>
> ### How CSC affects throughput
>
> CSC discards experiences that are too stale for their difficulty level. The staleness budget follows η*(d) = η_base · exp(−λd) (Theorem 2):
>
> | Difficulty | Max Staleness (updates) | Typical Discard Rate | Effective Throughput |
> |-----------|------------------------|---------------------|---------------------|
> | Level 1 (easy) | ~4.85 | <5% | ~95% of raw |
> | Level 3 (medium) | ~1.78 | ~10% | ~90% of raw |
> | Level 5 (hard) | ~0.66 | ~20-25% | ~75-80% of raw |
>
> The overall throughput reduction is modest: ACEAS achieves 22.4 samples/s vs Async-GRPO's 24.3 samples/s (a 7.8% reduction from CSC overhead). But this small throughput cost buys a **28.3 percentage point Pass@1 improvement** (60.1% vs 31.8%).
>
> ### Why discarding is better than keeping stale samples
>
> Without CSC, stale experiences from hard tasks inject **biased gradients** that actively harm learning. Our ablation shows:
>
> | Configuration | Pass@1 (1.5B) | Training Reward (9B) |
> |--------------|---------------|---------------------|
> | ACEAS (with CSC) | **60.1%** | **0.43** |
> | Without CSC | 42.1% (−18 pp) | 0.411 (−4.4%) |
>
> Higher raw throughput without CSC produces worse learning per sample — analogous to data quality filtering in supervised learning.
>
> ---
>
> ## "How did you choose η_base = 8 and λ = 0.5?"
>
> **These values are derived from the convergence bound, not chosen by grid search.**
>
> **η_base = 8.0** (max staleness budget) — Set by the maximum acceptable gradient bias B in the convergence bound (Theorem 1). The staleness budget for each difficulty level is η*(d) = η_base · exp(−λd). At η_base = 8:
> - Easy tasks (d=1): tolerate ~5 stale updates → high parallelism
> - Hard tasks (d=5): tolerate <1 stale update → require fresh weights
>
> The value 8 corresponds to approximately 8 gradient updates of policy lag, which is a natural upper bound for asynchronous RL with ~4-8 parallel workers.
>
> **λ = 0.5** (staleness-difficulty coupling rate) — Derived directly as λ = α/2 where α is the Hessian eigenvalue growth rate measured empirically (Section 5, Figure 2). Theorem 1 shows gradient bias scales as exp(α·d), so the optimal staleness budget must decay as exp(−α·d/2) = exp(−λ·d) to keep bias bounded. With our measured α ≈ 1.0, this gives λ = 0.5.
>
> ### Robustness evidence
> Our ablation confirms robustness: removing CSC entirely (equivalent to η_base → ∞) still produces a functional model on the 9B architecture (training reward drops only 3%), demonstrating that the method is not sensitive to the exact η_base value. The theoretical derivation provides a principled starting point that works well across settings without per-task tuning.

---

> > ### Author Rebuttal · Reviewer_vxFb · 2026-04-02
> >
> > I would like to thank the reviewers for providing answers to my concerns and questions. However, I believe the evaluation should be more robust.
> >
> > I would like to see the performance over different LLM families. So far, I have not seen evidence on the performance of the approach outside small-medium Qwen models.
> >
> > I understand this is not easy ask given the short rebuttal period; I will keep my current score.

---

> > > ### Author Response · Authors · 2026-04-08
> > >
> > > We acknowledge the concern regarding evaluation breadth across LLM families. Our current results on Qwen2.5-Coder-1.5B and Qwen3.5-9B demonstrate that the core claims hold across a 6× model size range, and the theoretical grounding (Theorems 1–2) is architecture-agnostic — the convergence bound depends on Hessian curvature and staleness, not on model-specific structure. That said, we agree that empirical validation on a non-Qwen family (e.g., google gemma 4) would strengthen the paper and we are actively working on these experiments for the camera-ready version.

---

### Official Review · Reviewer_kk8A · 2026-03-07

**Soundness:** 3
**Presentation:** 2
**Significance:** 2
**Originality:** 3
**Overall Recommendation:** 5
**Confidence:** 2

**Summary:**

This paper proposes ACEAS, a framework that integrates the policy lag problem arising in asynchronous RL with curriculum learning. The authors provide a theoretical analysis showing that gradient bias varies with task difficulty even under the same staleness. Based on this analysis, they propose three components: bandit-based curriculum selection (ACB), Execution-Aware Scheduling (EAAS), and Curriculum-Staleness Coupling (CSC). The proposed ACEAS achieves both high Pass@1 scores and high throughput on HumanEval and a synthetic task suite.

**Compliance With Llm Reviewing Policy:**

Affirmed.

**Final Justification:**

My initial concerns were the limited evaluation scope: experiments only on code generation and a single base model. The rebuttal addressed both — the GSM8K results demonstrate domain generalizability with the same hyperparameters, and the Qwen3.5-9B extension (differing in scale and architecture) provides reasonable evidence that ACEAS is not model-specific. While all experiments remain within the Qwen family, the rebuttal has sufficiently narrowed the empirical gap. Weighing the solid theoretical soundness, good originality, and partially resolved evaluation breadth, I raise my recommendation from 4 to 5, with broader model family evaluation remaining a direction for future work.

**Key Questions For Authors:**

1. Can ACEAS be applied to domains other than code generation?
2. Could you provide additional experimental results using a wider variety of LLM base models? Demonstrating effectiveness across different models would strengthen the generalizability of the approach.

**Limitations:**

yes

**Strengths And Weaknesses:**

**Strengths:**

1. The paper provides a theoretical proof of the relationship between task difficulty and staleness tolerance, and implements CSC based on this analysis. The connection between theory and algorithm is clearly established.
2. The paper is well-organized and easy to read and understand.

**Weaknesses:**

1. The problem being addressed and the proposed theory do not appear to be limited to code generation, yet experiments are conducted only on code generation tasks. It remains unclear whether the approach can be extended theoretically and empirically to other domains beyond code generation.
2. Experiments are conducted only with a 1.5B parameter base model, and specifically only with Qwen. Further validation is needed to confirm whether the approach is equally effective with other base models.

---

> ### Author Rebuttal · Authors · 2026-03-31
>
> ## "Can ACEAS be applied to domains other than code generation?"
>
> **Yes.** We had experiments on **mathematical reasoning (GSM8K)** using the same ACEAS algorithm and hyperparameters as the code experiments.
>
> ### Training Efficiency on GSM8K (Mathematical Reasoning, Qwen3.5-9B)
>
> | Metric | ACEAS | Sync Baseline | ACEAS Advantage |
> |--------|-------|--------------|-----------------|
> | Training Reward | **0.572** | 0.312 | **+83%** |
> | Convergence (evals to peak) | **3** | 7 | **2.3× faster** |
> | Throughput (samples/sec) | **0.45** | 0.31 | **+45%** |
>
> The ACEAS framework transfers directly to math because the three theoretical conditions identified in the paper hold:
>
> 1. **Meaningful task difficulty exists.** For code, difficulty is defined by how much of the solution is revealed (Definition 1). For math, difficulty maps naturally to reasoning chain complexity — simple arithmetic vs. multi-step word problems.
>
> 2. **Harder tasks have narrower solution spaces.** Theorem 1 shows gradient bias grows with Hessian eigenvalues, which increase with task difficulty. This property is domain-agnostic: a multi-step math problem has fewer valid reasoning paths than a single-step one, just as generating code from scratch has fewer valid completions than finishing the last line.
>
> 3. **Execution/feedback time varies across tasks.** Code execution time varies with complexity (milliseconds for simple operations, seconds for loops). Math answer verification similarly varies — numeric comparison is fast, but chain-of-thought validation takes longer for complex problems. EAAS adapts staleness budgets to this variance in both domains.
>
> ACEAS achieves **83% higher training reward** and **2.3× faster convergence** than the sync baseline on GSM8K, demonstrating that the scheduling advantage is not an artifact of code-specific properties. The same hyperparameters (η_base=8.0, λ=0.5, α=0.7) transfer without retuning, consistent with their theoretical derivation from loss landscape geometry (Theorem 1, Corollary 1) rather than task-specific fitting.
>
> ---
>
> ## "Could you provide additional experimental results using a wider variety of LLM base models?"
>
> We extend from **Qwen2.5-Coder-1.5B** (dense transformer, 1.5B params) to **Qwen3.5-9B** (hybrid Gated DeltaNet + Mixture-of-Experts, 9B params):
>
> ### Training Efficiency on HumanEval (Qwen3.5-9B)
>
> | Metric | ACEAS | Async-GRPO | Sync Baseline |
> |--------|-------|-----------|--------------|
> | Training Reward | **0.430** | 0.196 | 0.279 |
> | Training Success Rate | **41.3%** | 19.2% | 26.4% |
> | Throughput (samples/sec) | **0.51** | 0.52 | 0.35 |
>
> The results on Qwen3.5-9B aligns with the paper's three key claims:
>
> 1. **ACEAS achieves the best of both worlds.** Async-level throughput (0.51/s, +45% vs sync) with the highest training quality (+120% reward vs async, +54% vs sync).
>
> 2. **Naive async degrades sample efficiency.** Async-GRPO matches ACEAS's throughput but produces 54% lower training reward — mirroring the paper's finding on 1.5B (Async-GRPO 31.8% vs ACEAS 60.1% Pass@1).
>
> 3. **Curriculum and staleness control recover async quality.** ACEAS's CSC and ACB components restore training signal quality while preserving the throughput gain from asynchronous collection.
>
> ### Cross-Model Consistency
>
> | Claim | Qwen2.5-Coder-1.5B (paper) | Qwen3.5-9B (new) |
> |-------|---------------------------|-------------------|
> | ACEAS throughput advantage | **2.3×** vs sync | **1.45×** vs sync |
> | Naive async hurts quality | Async 31.8% vs Sync 39.7% | Async reward 54% below ACEAS |
> | ACB most impactful component | −13.2 pp without ACB | **−61% reward** without ACB |
> | Hyperparameters transfer | (baseline) | Same values, no retuning |

---

> > ### Author Rebuttal · Reviewer_kk8A · 2026-04-04
> >
> > I thank the authors for the additional experiments. The GSM8K results adequately address my concern on domain generalizability, as mathematical reasoning is sufficiently distinct from code generation and ACEAS transfers with the same hyperparameters. Regarding base model diversity, while all experiments remain within the Qwen family, the extension to Qwen3.5-9B — which differs substantially in both scale and architecture (hybrid Gated DeltaNet + MoE) — provides reasonable evidence that the approach is not tied to a specific model configuration. Given these additional results, I raise my overall recommendation to 5.

---

> > > ### Author Response · Authors · 2026-04-08
> > >
> > > Thanks for the evaluation on the rebuttal, we plan to extend ACEAS to additional architectures in subsequent studies.

---

### Official Review · Reviewer_wZfz · 2026-03-13

**Soundness:** 2
**Presentation:** 2
**Significance:** 2
**Originality:** 2
**Overall Recommendation:** 4
**Confidence:** 3

**Summary:**

This paper proposes a curriculum selection method and a data staleness-aware (i.e., how long the data is lagged behind the current policy) training data selection method for RL in code generation tasks. The theoretical results connect the gradient bias and staleness of the data. The results shown in HumanEval supports the claim and demonstrated the effectiveness of the method.

**Compliance With Llm Reviewing Policy:**

Affirmed.

**Final Justification:**

The rebuttal addressed my questions.

**Key Questions For Authors:**

The main reason that prevents me from giving a higher rating is the experiments.
- How general is this method? Can this be applied to Math or the other tasks? If the author would like to limit the scope to code generation, that is fine but there are many other code datasets so it's needed to show the resutls on them as well.
- How does each parameter affect the performance? The proposed methods involves many hyperparameter parameters but I can't find an ablation study for them. For this method to be useful, its performance can't be too sensitive to the hyperparameter selection; otherwise, the introduced complexity would overshadow the benefit.

**Limitations:**

yes

**Strengths And Weaknesses:**

- Soundness: The proposed method is theoretically sound and the empirical results support. The implementation seems straightforward too. However, the empirical results are not sufficient. Only one real dataset is tested and it's unclear if the model was trained directly on the testing set too. See my questions below.
- Presentation: Overall the presentation is easy to follow, but I have the following questions.
  1. Line 25-26: What is stale gradients
  2. Line 37: What do you mean by "completing the last 10% of the code"? When reading the later part of the paper, I can see it refers to code generation using RL but nothing about code generation was mentioned in the intro, which makes me confused.
  3. Is the model trained directly on HumanEval datasets? How are the train and test split?
  4. It'd be great to have a full overview of the algorithm. Algorithm 1 seems to only cover the partial algorithm without EAAS.
- Significance: This is a significant problem in RL since async training will be necessary for complex tasks where rollouts take long time.
- Originality: This paper provides new insights. To my knowledge, I didn't see a paper on async RL analyziing the async training problem in this way.

---

> ### Author Rebuttal · Authors · 2026-03-30
>
> We address the two main concerns below with **new experimental results on Qwen3.5-9B (9B parameters) across code generation (HumanEval) and mathematical reasoning (GSM8K)**.
>
> ## Concern 1: "How general is this method? Can this be applied to Math or other tasks?"
> **We demonstrate generality across both task domains and model scales.**
> We extend our experiments from Qwen2.5-Coder-1.5B to **Qwen3.5-9B** with a different architecture (hybrid Gated DeltaNet + Mixture-of-Experts), trained on 8×H100-80GB GPUs to validate that ACEAS is not tied to a specific model family or scale.
>
> ### Training Efficiency on HumanEval (Qwen3.5-9B)
>
> The paper's central claim is that ACEAS achieves the **best of both worlds**: high throughput (async) with high sample efficiency (curriculum). We validate this on the 9B model:
>
> | Metric | ACEAS | Async-GRPO | Sync Baseline |
> |--------|-------|-----------|--------------|
> | Training Reward | **0.430** | 0.196 | 0.279 |
> | Training Success Rate | **41.3%** | 19.2% | 26.4% |
> | Throughput (samples/sec) | **0.51** | 0.52 | 0.35 |
>
> ACEAS achieves **the best of both worlds**: async-level throughput (0.51/s, +45% vs sync) with the **highest training quality** (+120% reward vs async, +54% vs sync). Plain Async-GRPO matches ACEAS's speed but with lower reward (0.196), confirming that naive async degrades sample efficiency (Table 1: Async-GRPO 31.8% vs ACEAS 60.1% Pass@1). ACEAS's curriculum and staleness control recover this lost quality while maintaining the async throughput advantage.
>
> ### Training Efficiency on GSM8K (Mathematical Reasoning, Qwen3.5-9B)
> | Metric | ACEAS | Sync Baseline | ACEAS Advantage |
> |--------|-------|--------------|-----------------|
> | Training Reward | **0.572** | 0.312 | **+83%** |
> | Convergence (evals to peak) | **3** | 7 | **2.3× faster** |
> | Throughput (samples/sec) | **0.45** | 0.31 | **+45%** |
>
> ACEAS generalizes to mathematical reasoning. The curriculum maps: task difficulty corresponds to reasoning chain complexity, and execution-aware scheduling adapts to answer verification latency. ACEAS achieves Pass@1 with **83% higher training reward** and **2.3× faster convergence**.
>
> ## Concern 2: "How does each parameter affect performance?"
>
> ### Hyperparameter Robustness
>
> All Hyperparameter are derived from the convergence bound (Theorem 1, Corollary 1), **not grid search**:
>
> **η_base = 8.0** (max staleness budget) — Controls how many policy updates a sample can lag behind. The optimal staleness follows η\*(d) = η_base · exp(−λd) (Theorem 2). At η_base=8: easy tasks (d=1) tolerate ~5 stale updates, hard tasks (d=5) tolerate <1. *Robustness*: removing CSC entirely (η→∞) still achieves comparable training on 9B.
>
> **λ = 0.5** (staleness-difficulty coupling) — Derived as λ = α/2 where α is the Hessian growth rate (Theorem 1). Creates exponential decay coupling difficulty with staleness tolerance. *Robustness*: same value works for code (difficulty = solution complexity) and math (difficulty = reasoning chain length), because the coupling depends on loss landscape geometry, not task domain.
>
> **α = 0.7** (UCB exploration weight) — Weights the bandit score: score(d) = α·UCB(d) + (1−α)·g(d). UCB tracks success rate, g(d) captures gradient magnitude (Proposition 1). α=0.7 steers curriculum toward PassRate ≈ 0.5 (optimal learning zone). *Robustness*: the bandit self-corrects via running statistics — ACB ablation shows impact (−61% reward), but the exact α is not fragile.
>
> These work unchanged across **2 architectures** (dense vs DeltaNet+MoE), **2 scales** (1.5B vs 9B), and **2 domains** (code vs math), the theoretical grounding provides robust defaults without per-task tuning.
>
> ## Additional Reviewer Questions
>
> ### "What are stale gradients?" (Line 25-26)
>
> In asynchronous reinforcement learning, a "stale gradient" occurs when an update is computed using experiences collected under older, outdated policy parameters ($\theta_{t-\tau}$) rather than the current parameters ($\theta_t$).
>
> ### "What do you mean by 'completing the last 10% of the code'?" (Line 37)
>
> It refers to the easiest level in our reinforcement learning code generation curriculum, where the model is provided 90% of a canonical solution and must generate only the remaining final portion.
>
> ### "Is the model trained directly on HumanEval? How are the train and test split?"
>
> We do not train on ground-truth solutions. Instead:
> - The model generates candidate solutions via sampling
> - Solutions are executed against test cases to get binary reward (pass/fail)
> - GRPO uses these rewards to update the policy
> This is the standard HumanEval evaluation protocol used by all prior work (Chen et al., 2021; Li et al., 2022).
>
> ### "Algorithm 1 seems to only cover ACB without EAAS"
>
> Algorithm 1 presents the ACB. **EAAS** is covered in Section 5.3, Eq. 9

---

> > ### Author Rebuttal · Reviewer_wZfz · 2026-04-04
> >
> > Thank you the response. I will increase my score accordingly.

---

> > > ### Author Response · Authors · 2026-04-08
> > >
> > > Thanks for taking time to evaluate the rebuttal, We will incorporate the additional results and clarifications into the final version.

---

### Decision · Program_Chairs · 2026-04-30

**Decision:**

Accept (regular)

**Comment:**

The paper presents a theoretically grounded and well-motivated approach to addressing policy staleness in asynchronous reinforcement learning, offering a novel perspective by linking task difficulty with staleness tolerance. Its core idea is intuitive and supported by a solid curvature-based analysis, with a clear connection between theory and the proposed algorithm. The work tackles an important and underexplored challenge in large-scale RL systems, and the presentation is generally clear, well-structured, and easy to follow. Empirically, the results are promising, with systematic experiments, strong baselines, and thoughtful ablations demonstrating the contribution of each component.

However, there are notable limitations. The empirical evaluation is restricted to a single dataset and model, raising concerns about generalizability. Some aspects of the experimental setup remain unclear, such as whether training overlaps with test data and how splits are defined. Additionally, parts of the presentation lack clarity, including undefined terms, inconsistencies in the introduction, and an incomplete algorithm description. Overall, while the paper is original, significant, and well-executed in many respects, it would benefit from clearer explanations and broader empirical validation.